# Quantile Rendering: Efficiently Embedding Language Features on 3D Gaussian Splatting

## Abstract

Recent advancements in computer vision have successfully extended Open-vocabulary segmentation (OVS) to the 3D domain by leveraging 3D Gaussian Splatting (3D-GS). Despite this progress, efficiently rendering the high-dimensional features required for open-vocabulary queries poses a significant challenge. Existing methods employ codebooks or feature compression, causing information loss, thereby degrading segmentation quality. To address this limitation, we introduce Quantile Rendering (Q-Render), a novel rendering strategy for 3D Gaussians that efficiently handles high-dimensional features while maintaining high fidelity. Unlike conventional volume rendering, which densely samples all 3D Gaussians intersecting each ray, Q-Render sparsely samples only those with dominant influence along the ray. By integrating Q-Render into a generalizable 3D neural network, we also propose Gaussian Splatting Network (GS-Net), which predicts Gaussian features in a generalizable manner. Extensive experiments on ScanNet and LeRF demonstrate that our framework outperforms state-of-the-art methods, while enabling real-time rendering with an approximate $\sim 43.7\times$ speedup on 512-D feature maps. Code will be made publicly available.

## 1 Introduction

3D Gaussian Splatting (3D-GS) (Kerbl et al., 2023) has emerged as a powerful representation for the neural rendering task, offering an explicit set of 3D Gaussians combined with efficient splatting (Zwicker et al., 2001) and tile-based rasterization for real-time rendering. This pipeline delivers both high-quality reconstruction and real-time frame rates—capabilities that have quickly made it a foundation for numerous 3D vision and graphics applications. Beyond photorealistic rendering, recent works (Qin et al., 2024; Wu et al., 2024b; Ye et al., 2024; Lyu et al., 2024; Jun-Seong et al., 2025) have begun leveraging 3D Gaussians as a medium for scene understanding. A prevalent approach is to distill knowledge from powerful 2D foundation models—such as CLIP (Radford et al., 2021; Ilharco et al., 2021), SAM (Kirillov et al., 2023; Ravi et al., 2024), and DINO (Caron et al., 2021; Oquab et al., 2023; Siméoni et al., 2025)—into pre-optimized 3D Gaussians by embedding high-dimensional features. In this work, we focus on open-vocabulary segmentation (OVS) using OpenCLIP (Ilharco et al., 2021), which outputs 512-dimensional language-aligned features for arbitrary text or image queries.

The original volume rendering algorithm (Kerbl et al., 2023) samples *all* Gaussians intersecting a ray, regardless of their actual contribution to the output, *i.e.* rendered pixel color. For RGB rendering this overhead is manageable, but for high-dimensional embeddings (e.g., 512-D CLIP features) it becomes computationally heavy. To resolve this problem, series of studies compress the dimensionality of 512-D CLIP features into 3-D or 6-D features or codebooks (Qin et al., 2024; Wu et al., 2024b; Zhou et al., 2024; Jun-Seong et al., 2025). While effective, this strategy is not a fundamental solution and can potentially loss the original information that was stored in the high-dimensional features. Moreover, the distribution of the optimized 3D Gaussians potentially have noisy or local minima due to its per-scene optimization scheme. Accordingly, it is challenging to properly embed high-dimensional feature vectors on the top of these 3D Gaussians.

Based on this observation, we hypothesize that not all Gaussians are influential—only a partial fraction of 3D Gaussians meaningfully affect the high-dimensional feature rendering along a ray. This observation motivates Quantile Rendering (Q-Render), our transmittance-aware and efficient

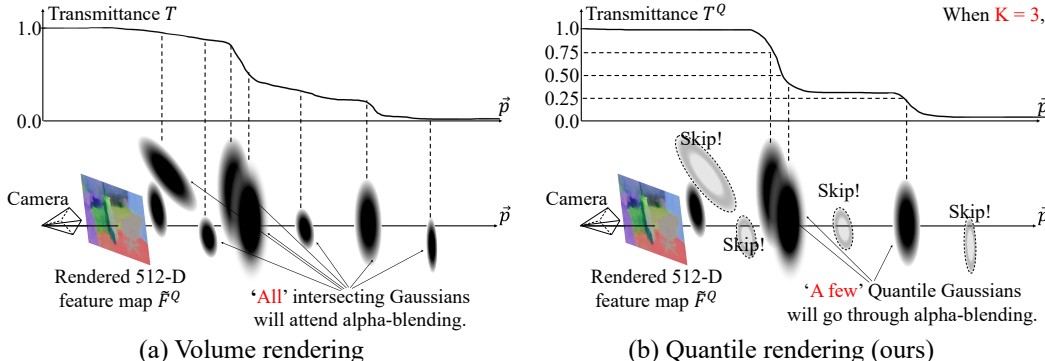

Figure 1: Quantile rendering. (a) Unlike volume rendering (Mildenhall et al., 2021; Kerbl et al., 2023) that densely samples and blends all 3D Gaussians along the rays, (b) our Quantile render selectively samples and blends a sparse set of quantile Gaussians – those with dominant influence along the ray, which can efficiently render high-dimensional feature maps from Gaussian features.

rendering algorithm for high-dimensional Gaussian features. Instead of densely accumulating every rasterized 3D Gaussian, Q-Render adaptively selects a small set of *quantile Gaussians*—those that dominate the ray's transmittance profile—and renders only these representatives. This quantile-based selection cuts redundant computation, and approximates the original signals for downstream tasks that may require to render high-dimensional feature maps.

In this work, integrating Q-Render into a 3D neural network (Choy et al., 2019; Wu et al., 2024a; Chen et al., 2024), we build Gaussian Splatting Network (GS-Net) that operates on 3D Gaussians to predict Gaussian features. Typically, Q-Render serves as an efficient bridge between 2D super-vision and the 3D neural network, allowing backward gradients (Rumelhart et al., 1986) to flow from image-space losses to the 3D neural network's predictions. Moreover, with this integration, Q-Render's sparse sampling becomes even more advantageous: the inductive bias of the 3D neural network tends to predict spatially smooth Gaussian features, meaning that densely sampling all Gaussians along each ray is unnecessary. Instead, the sparsely selected quantile Gaussians are sufficient to faithfully render high-dimensional feature maps while significantly reducing computational overhead during the rendering process and its backward computation.

We validate our method on two open-vocabulary 3D semantic segmentation benchmarks: (1) Scan-Net and (2) LeRF-OVS, where CLIP-based embeddings are stored directly in 3D Gaussians. In both cases, Q-Render achieves state-of-the-art results, underscoring its value as a scalable bridge between 2D foundation models and 3D Gaussian representations.

- *Quantile Rendering* — a sparse, transmittance-guided sampling strategy that selects only the most representative Gaussians along each ray for efficiency.

- *Gaussian Splatting Network* — A 3D neural network that predicts high-dimensional fea-turemetric Gaussians from optimized 3D Gaussians, with Q-Render enabling efficient and effective feature distillation from 2D foundation models.

- *Extensive Validation* — comprehensive experiments on open-vocabulary 3D semantic seg-mentation benchmarks with superior performances against recent studies.

## 2 RELATED WORKS

**High-dimensional Features on Gaussian representation.** There is a growing body of this field that adopts 3D Gaussian Splatting (Kerbl et al., 2023) as the underlying representation, incorporat-ing these language-aligned embeddings into per-Gaussian memory (Qin et al., 2024; Shi et al., 2024; Zhou et al., 2024; Wu et al., 2024b; Jun-Seong et al., 2025). The core idea is to utilize a render-ing pipeline of vanilla 3DGS (Kerbl et al., 2023) to sample 3D Gaussians for distillation from 2D image embeddings, a capability that point clouds inherently lack. Nonetheless, their memory-based approaches force per-scene feature optimization, which limits their application in practice. More-

over, lifting high-dimensional embeddings (*e.g.*, CLIP 512-D feature) into 3D Gaussians demands significant computational resources.

**Neural scene representation networks.** Recent research has explored treating neural scene representations as 3D models, using neural networks to process them and solve various 3D tasks. PeRFception (Jeong et al., 2022) trains networks to process Plenoxels (Fridovich-Keil et al., 2022) for classification and semantic segmentation. SPARF (Hamdi et al., 2023) improves Plenoxel in a few-shot setup by training networks that process few-shot Plenoxels and output the enhanced version. SplatFormer (Chen et al., 2024) adopts Point Transformer V3 (Wu et al., 2024a) to improve the robustness of 3D-GS under out-of-distribution poses. To the best of our knowledge, we are the first to address language and grouping tasks using networks that process 3D-GS.

## 3 Preliminary

**3D Gaussian Splatting.** 3D-GS (Kerbl et al., 2023) represents 3D scenes as a set of 3D Gaussians where each 3D Gaussian $\mathbf{g}$ is parameterized as center position $\boldsymbol{\mu} \in \mathbb{R}^3$, covariance matrix $\boldsymbol{\Sigma} \in \mathbb{R}^{3\times3}$, opacity $\alpha \in \mathbb{R}$, and spherical harmonics coefficients with $d$ degree $\mathbf{sph} \in \mathbb{R}^{(d+1)^2\times3}$. In particular, the covariance matrix of the anisotropic Gaussian distribution is decomposed into scaling factors $\mathbf{s} \in \mathbb{R}^3_{>0}$, rotation in quaternion representation $\mathbf{r} \in \mathbb{R}^4$, where $\boldsymbol{\Sigma} = \mathbf{R}\mathbf{S}\mathbf{S}^\top\mathbf{R}^\top$. Given a set of $N$ numbers of 3D Gaussians $\mathcal{G} = \{\mathbf{g}_i\}_{i=1}^N = \{\boldsymbol{\mu}_i, \mathbf{s}_i, \mathbf{r}_i, \alpha_i, \mathbf{sph}_i\}_{i=1}^N$, 3D-GS (Kerbl et al., 2023) organizes the rendering pipeline as: (1) tiling strategy; (2) splatting algorithm (Zwicker et al., 2001); and (3) volume rendering (Mildenhall et al., 2021). In details, volume rendering proceeds with alpha-blending through densely rasterized 3D Gaussians along a ray $\overrightarrow{p}$ as:

$$\tilde{\mathbf{C}}[\overrightarrow{p}] = \sum_{i\in S_r} w_i\mathbf{c}_i \quad \text{s.t.} \quad w_i = T_i\alpha_i', \;\; T_i = \prod_{j\in S_{r,:i-1}}(1-\alpha_j'), \;\; \alpha_i' = \alpha_i \cdot \mathrm{G}_{\boldsymbol{\mu}_i,\boldsymbol{\Sigma}_i}(u) \tag{1}$$

where $\tilde{\mathbf{C}}$ is a rendered image, $\tilde{\mathbf{C}}[\overrightarrow{p}]$ is a rendered pixel color at the ray $\overrightarrow{p}$, $\mathbf{c}_i$ is the emitted color of the $i$-th Gaussians obtained from $\mathbf{sph}_i$, $S_r$ is an ordered sequence of indices of densely sampled Gaussians along $\overrightarrow{p}$, $T_i$ is the transmittance at $i$-th Gaussian along $\overrightarrow{p}$, and $\mathrm{G}(\cdot)$ is the Gaussian function evaluated at pixel location $u$ by projecting the Gaussian parameters (Eq. **??** of the appendix). For training, 3D-GS optimizes parameters $\mathcal{G}$ by minimizing a rendering loss $\mathcal{L}_{\mathrm{rgb}} = \|\mathbf{C} - \tilde{\mathbf{C}}\|_1 + \lambda_{\mathrm{SSIM}} \cdot \mathrm{SSIM}(\mathbf{C}, \tilde{\mathbf{C}})$ where $\mathbf{C}$ is a ground truth image and $\lambda_{\mathrm{SSIM}}$ is the hyperparameter adjusting the SSIM loss.

**High-dimensional Gaussian features.** Recent studies (Qin et al., 2024; Wu et al., 2024b; Ye et al., 2024; Lyu et al., 2024; Zhou et al., 2024) propose to register high-dimensional features for each 3D Gaussian $\mathbf{g}$. These embedded features store 3D scene information such as language attributes, mask identities, *etc*. These methods commonly start from optimized 3D Gaussians $\mathcal{G}$ following the original 3D-GS paper (Kerbl et al., 2023). Then, while freezing $\mathcal{G}$, this method allocates $C$-dimensional feature vector $\mathbf{f} \in \mathbb{R}^C$ for every 3D Gaussian $\mathbf{g}$ such that these feature vectors are optimized to minimize feature rendering losses.

## 4 Methodology

As shown in Figure 2, given a set of $N$ numbers of optimized 3D Gaussians $\mathcal{G} = \{\mathbf{g}_i\}_{i=1}^N$, a 3D neural network predicts a set of $C$-dimensional Gaussian features $\mathcal{F} = \{\mathbf{f}_i\}_{i=1}^N$ where $\mathbf{f} \in \mathbb{R}^C$. Through our Quantile Rendering, the Gaussian features are rendered into $C$-dimensional feature maps. The 3D neural network is trained to minimize the rendering loss between the rendered feature maps and the target 2D feature maps extracted from CLIP's vision encoder.

### 4.1 3D neural network

Conventional 3D neural networks for pointcloud (Choy et al., 2019; Wu et al., 2024a; Choe et al., 2021; Yang et al., 2024; Ding et al., 2023; Lee et al., 2025) often voxelize 3D points to efficiently process scene-scale 3D points. They transform scattered points into sparse voxel grids with unique

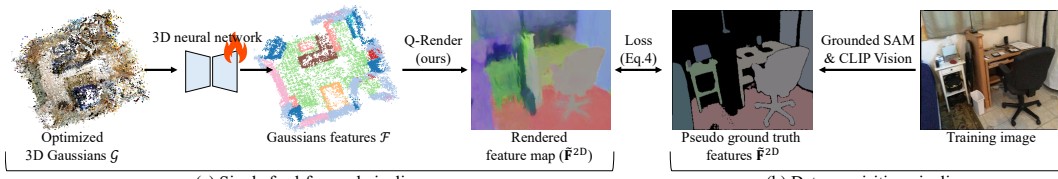

Figure 2: Overview. Given optimized 3D Gaussians $\mathcal{G}$, our network is trained to predict Gaussian features $\mathcal{F}$ that are aligned with the language embedding space from CLIP's vision encoder. Typically, the proposed Q-Render accelerates the training and inference speed by transforming predicted Gaussian features into rendered feature maps.

spatial locations, enabling single-pass inference over an entire 3D scene thanks to its efficiency. In contrast to handling pointclouds, modeling a scene with continuous 3D Gaussians (Kerbl et al., 2023) introduces overlapping regions since each Gaussian has a volumetric extent. To resolve this issue, we follow SplatFormer (Chen et al., 2024) that introduces voxelization on 3D Gaussians by sampling its center location $\boldsymbol{\mu}$ to ensure compatibility with typical 3D backbones: Point Transformer v3 (PTv3) (Wu et al., 2024a) and MinkUnet (Choy et al., 2019). These models are designed to predict voxel features such that we proceed with the de-voxelization steps to convert predicted voxel features into predicted Gaussian features $\mathcal{F}$, which will be used for rendering procedures.

## 4.2 QUANTILE RENDERING

To train the 3D neural network using the knowledge from 2D foundation models, rendering process is necessary. Volume rendering (Mildenhall et al., 2021; Kerbl et al., 2023) can be an option, but it may require large computation power (Qin et al., 2024; Wu et al., 2024b; Zhou et al., 2024) when rendering high-dimensional Gaussian features. This is because volume rendering repeatedly accumulates high-dimensional Gaussian features $\mathcal{F}$ along the ray, which quickly becomes prohibitively expensive as dimensionality grows as described in Table 1.

To address this, we introduce a quantile-based sampling strategy. Quantiles—statistical cutpoints that divide a distribution into intervals of equal probability—enable sparse yet representative sampling. By approximating the distribution in traditional volume rendering with a carefully selected subset of Gaussians, our Quantile Rendering achieves efficient rendering while preserving representativeness. Specifically, given a hyperparameter $K$, the algorithm samples $K$ Quantile Gaussians along each ray. The detailed procedure is presented in Algorithm 1. We first rasterize and splat 3D Gaussians following (Kerbl et al., 2023) to obtain the indices of rasterized 3D Gaussians $I$ at the target ray $\overrightarrow{p}$ as described in Section 3. Then, our Q-Render takes place to run three sub-steps: Quantile Gaussian sampling, alpha-blending, and feature normalization.

**Sampling Quantile Gaussian.** Given hyperparameter $K$, we partition the transmittance $T \in \mathbb{R}_{[0,1]}$ into $K+1$ evenly distributed segments at each ray, and track how much the transmittance changes by passing through each Gaussian. Once crossing a segment boundary( line 6 of Algorithm 1), we determine that this Gaussian as Quantile in this interval.

**Alpha-blending on Quantile Gaussians.** With these Quantile Gaussians that show the relatively big transmittance change, we participate such Gaussians in alpha-blending. As summarized in Table 1, this selective blending reduces the time complexity from $\mathcal{O}(NC)$ (volume rendering) to $\mathcal{O}(N + KC)$, where $N$ is the number of Gaussians for each pixel and $C$ is the feature dimension. Unlike top-$K$ sampling proposed by a concurrent study (Jun-Seong et al., 2025), which requires additional complexity to sort values, $\mathcal{O}(N \log K + KC)$.

**Normalizing feature vector.** In the volume rendering (Kerbl et al., 2023; Mildenhall et al., 2021), transmittance $T$ is initialized as one and then the remaining transmittance $T$ closes to zero as proceeding with the alpha blending along a ray. However, alpha-blending on sub-sampled Gaussians may have remaining transmittance that has relatively higher than 0. To approximate the original distribution in volume rendering that returns zero-close final transmittance value, we forcefully set the final transmittance $T^Q$ as zero by normalizing the accumulated feature $\tilde{\mathbf{f}}^Q$ as $\tilde{\mathbf{f}}^Q \leftarrow \frac{\mathbf{f}^Q}{1-T^Q}$.

---

**Algorithm 1** Quantile Rendering

---

**Require:** Optimized 3D Gaussians $\mathcal{G}$, predicted Gaussian features $\mathcal{F}$, the number of Quantile Gaussians $K$, indices of rasterized 3D Gaussians $I$ at target ray.

**Ensure:** Rendered feature vector at the target ray $\tilde{\mathbf{f}}^Q \in \mathbb{R}^C$.

1: **procedure** QUANTILERENDER($\mathcal{G}, \mathcal{F}, K, I$)
2:      $T \leftarrow 1, \ T^Q \leftarrow 1, \ \mathbf{f}^Q \leftarrow \mathbf{0}, \ k \leftarrow 0$
3:      **for** $i$ in $I$ **do**
4:          $\mathbf{g}_i \leftarrow \mathcal{G}[i], \mathbf{f}_i \leftarrow \mathcal{F}[i]$                            $\triangleright \ \mathbf{g}_i = \{\boldsymbol{\mu}_i, \mathbf{s}_i, \mathbf{r}_i, \alpha_i, \mathbf{sph}_i\}$
5:          $T_{\text{test}} \leftarrow T \cdot (1 - \alpha'_i)$
6:          **if** $T_{\text{test}} < 1 - \frac{k+1}{K+1}$ **then**                  $\triangleright$ Sampling Quantile Gaussian
7:             $k \leftarrow k + 1$
8:             $w^Q \leftarrow T^Q \cdot \alpha'_i$
9:             $\mathbf{f}^Q \leftarrow \mathbf{f}^Q + w^Q \cdot \mathbf{f}_i$               $\triangleright$ Alpha-blending on Quantile Gaussians
10:           $T^Q \leftarrow T^Q \cdot (1 - \alpha'_i)$
11:           **while** $T_{\text{test}} < 1 - \frac{k+1}{K+1}$ **do**
12:               $k \leftarrow k + 1$
13:           **end while**
14:          **end if**
15:          **if** $T_{\text{test}} < \frac{1}{K+1}$ **then**
16:            **break**
17:          **end if**
18:          $T \leftarrow T_{\text{test}}$
19:      **end for**
20:      $\tilde{\mathbf{f}}^Q \leftarrow \frac{\mathbf{f}^Q}{1 - T^Q}$                         $\triangleright$ Normalizing feature vector
21:      **return** $\tilde{\mathbf{f}}^Q$
22: **end procedure**

---

Table 1: Complexity comparison table. $K$ intervals along each ray, $N$ is the number of Gaussians per pixel and $C$ is the feature dimension. It shows the efficiency of our Q-Render, which avoids any multiplier for the large $N$. Precise inference time is in Figure 6.

| | V-Render (Kerbl et al., 2023) | top-$K$ (Jun-Seong et al., 2025) | Q-Render (ours) |
|---|---|---|---|
| Complexity | $\mathcal{O}(NC)$ | $\mathcal{O}(N \log K + KC)$ | $\mathcal{O}(N + KC)$ |

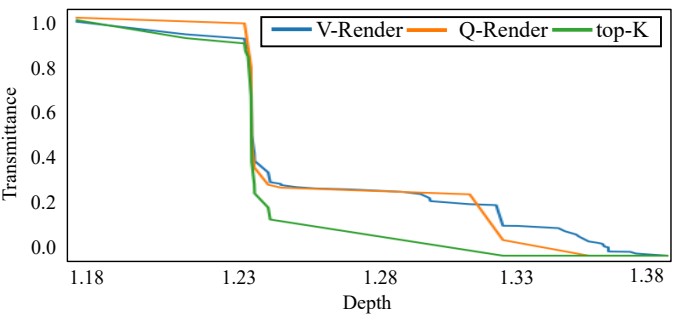

Figure 3: Comparison of transmittance distribution across different Gaussians sampling algorithms. Our Q-Render effectively approximates the distribution of the transmittance distribution of the original 3D-GS. We used $K = 10$ for visualization.

Finally, our Quantile rendering has been formulated as $\tilde{\mathbf{f}}^Q = \text{QuantileRender}(\mathcal{G}, \mathcal{F}, K, I)$ where $I$ is the indices of rasterized 3D Gaussians. As illustrated in Figure 3, Q-Render well approximates the transmittance tendency from the volume rendering (Kerbl et al., 2023) while top-$K$ sampling strategy (Jun-Seong et al., 2025) shows different the tendency. In short, our method avoids the overhead and outperforms top-$K$ in both efficiency and accuracy as stated in Table 6.

Table 2: Open vocabulary 3D semantic segmentation performances in the ScanNet dataset.

| Method | Per-scene optim. | 19 classes | | 15 classes | | 10 classes | |
|---|---|---|---|---|---|---|---|
| | | mIoU (↑) | mAcc (↑) | mIoU (↑) | mAcc (↑) | mIoU (↑) | mAcc (↑) |
| LangSplat (Qin et al., 2024) | ✓ | 1.47 | 10.23 | 2.00 | 11.85 | 3.24 | 16.79 |
| OpenGaussian (Wu et al., 2024b) | ✓ | 22.60 | 34.41 | 24.21 | 37.58 | 34.74 | 51.58 |
| Dr.Splat (Jun-Seong et al., 2025) | | 23.21 | 35.42 | 25.33 | 34.64 | 36.71 | 53.29 |
| GS-Mink (ours) | | **50.75** | **62.00** | **53.54** | **66.39** | **64.95** | **79.34** |
| GS-PTv3 (ours) | | 48.99 | 60.36 | 52.39 | 66.05 | 62.57 | 77.70 |

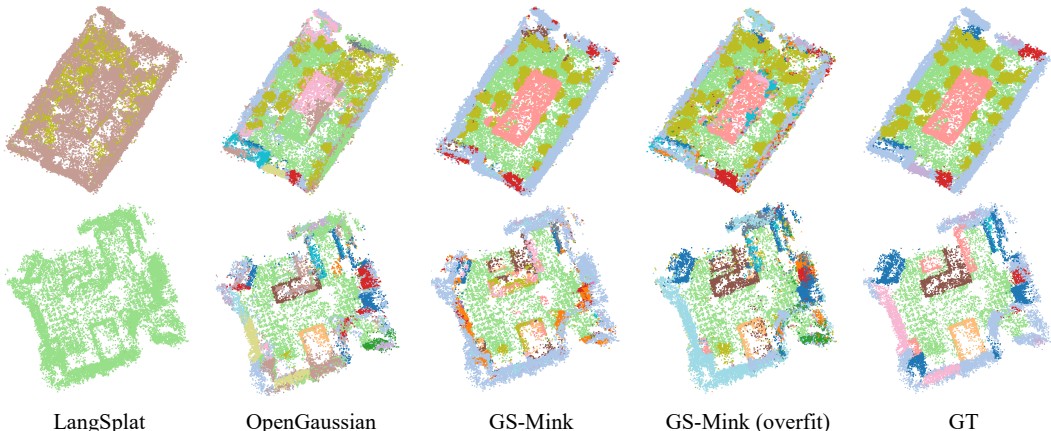

| LangSplat | OpenGaussian | GS-Mink | GS-Mink (overfit) | GT |
|---|---|---|---|---|

Figure 4: Qualitative results on the open-vocabulary 3D semantic segmentation task.

Furthermore, the inductive bias of the 3D neural network promotes spatially smooth Gaussian feature predictions. Accordingly rendering dense sampling along rays becomes redundant. Our sparse quantile selection remains sufficient for high-fidelity feature mapping while significantly reducing computational overhead during both rendering and backward passes. Such an approximation is bounded by $O(1/K)$ where we provide a detailed theoretical justification of Q-Render as a Riemann sum approximation in Appendix C.

## 4.3 TRAINING LOSS

We validate our GS-Net in open-vocabulary 3D semantic segmentation task with two datasets, ScanNet dataset and LeRF-OVS dataset. In this task, CLIP (Radford et al., 2021; Ilharco et al., 2021) is one of the popular vision language models that predicts a 512-D feature vector from a given input image. Following the previous studies (Qin et al., 2024; Wu et al., 2024b; Jun-Seong et al., 2025), we set the distillation target as CLIP vision encoder's feature vectors. We first extract image patches, *a.k.a.* masks $\{\mathbf{m}\}$, using Grounded-SAM2 (Ren et al., 2024), and extract CLIP embeddings from the corresponding patch $\mathbf{f}^{\text{CLIP}}$. Given a pair of training data $\{\mathbf{m}_i, \mathbf{f}_i^{\text{CLIP}}\}$, the 3D neural network is trained to minimize a discrepancy between our rendered feature vector $\tilde{\mathbf{f}}^Q$ and CLIP embedding $\mathbf{f}^{\text{CLIP}}$ in the format of the contrastive loss $\mathcal{L}$ as follows:

$$\mathcal{L} = -\log \frac{\exp(\text{sim}(\tilde{\mathbf{f}}^Q, \mathbf{f}_i^{\text{CLIP}}))}{\sum_{i \neq j} \exp(\text{sim}(\tilde{\mathbf{f}}^Q, \mathbf{f}_j^{\text{CLIP}}))}, \quad (2)$$

where $\text{sim}(\cdot, \cdot)$ is the cosine similarity function.

## 5 EXPERIMENTS

We provide experiments to verify the efficacy of our GS-Net on open-vocabulary 3D semantic segmentation task using the ScanNet dataset and LeRF-OVS dataset. Given a set of open-vocabulary text queries, we extract their text features using CLIP's text encoder. For fair comparison, we used

Table 3: Open-vocabulary semantic segmentation performances in the LeRF-OVS dataset.

| Method | Feature dim. | mIoU (↑) | mAcc (↑) |
|---|---|---|---|
| LangSplat (Qin et al., 2024) | 3 | 9.7 | 12.4 |
| LEGaussians (Shi et al., 2024) | 8 | 16.2 | 23.8 |
| OpenGaussian (Wu et al., 2024b) | 6 | 38.4 | 51.4 |
| SuperGSeg (Liang et al., 2024) | 64 | 35.9 | 52.0 |
| GS-Mink (ours) | 6 | 38.6 | 52.3 |
| GS-Mink (ours) | 512 | **45.8** | **56.9** |

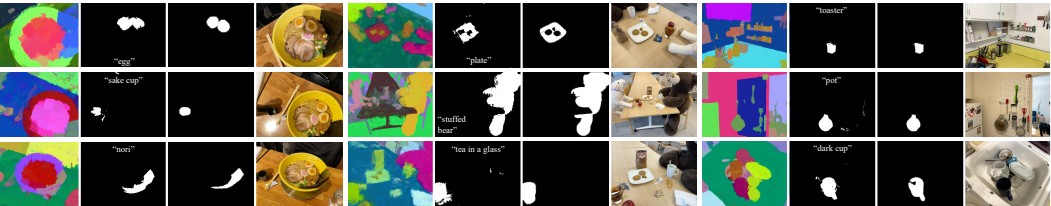

Figure 5: Our qualitative results in the LeRF-OVS dataset.

the sam CLIP model provided by the recent work (Jun-Seong et al., 2025). We then compute the cosine similarity between Gaussian features and these text features, assigning each Gaussian the labels with the highest similarity scores.

Following the evaluation protocol of OpenGaussian (Kerbl et al., 2023), we assess mIoU and mAcc on predefined sets of 19, 15, and 10 categories by theirs. In contrast to OpenGaussian, which freezes the original pointclouds and disables densification and pruning, we enable both densification and pruning to fully leverage the capacity of 3D-GS. As a result, per-Gaussian labels are required for evaluation that is extracted from pointcloud labels. To assign semantic labels to each Gaussian, we use the Mahalanobis distance to find the $K$ nearest neighbors, similar to Dr.Splat (Jun-Seong et al., 2025). Furthermore, we found that filtering points based on opacity and Gaussian influence leads to more reliable labels, as reported in section B.2. For a fair comparison, we reproduce the results of LangSplat (Qin et al., 2024), OpenGaussian (Wu et al., 2024b), and Dr.Splat (Jun-Seong et al., 2025) using our training and evaluation setup.

## 5.1 SCANNET DATASET

We compare GS-Net against baselines (Qin et al., 2024; Wu et al., 2024b) on the ScanNetv2 (Dai et al., 2017) dataset, which includes 1,513 scenes and 124,505 frames of indoor captures featuring various household objects. When training our generalized network, we use the same 10 validation scenes as OpenGaussian (Wu et al., 2024b) and train on the remaining 1,503 scenes. Additionally, we train GS-Net to overfit a single scene to ensure a fair comparison with the baselines. We pre-optimize all 3D-GS (Kerbl et al., 2023) using the original implementation across all scenes to train and evaluate GS-Net.

In Table 2, we compare two GS-Net models, GS-Mink and GS-PTv3 where each model has MinkUnet (Choy et al., 2019) and PTv3 (Wu et al., 2024a) as baseline 3D neural networks, against previous baselines (Qin et al., 2024; Wu et al., 2024b; Jun-Seong et al., 2025). Although the evaluation scenes are not used for training GS-Net, GS-Net achieves significant performance improvements or comparable results to previous baselines. Furthermore, when overfitted to a single scene, GS-Mink and GS-PTv3 improve mIoU by $12.08\%p$ and $12.73\%p$, respectively. Interestingly, while GS-PTv3 outperforms GS-Mink in the overfitting scenario, this trend reverses when training for generalization. We found that GS-PTv3 is more prone to overfitting during training, suggesting that architectural improvements for handling Gaussians could mitigate this issue. As shown in Figure 4, our GS-Net produces much clearer semantic segmentation results, further demonstrating the superiority of our method.

| 3D neural network | Feature renderer | $K$ | 19 classes mIoU ($\uparrow$) |
|---|---|---|---|
| | V-Render (Kerbl et al., 2023) | - | 49.02 |
| MinkUNet | top-$K$ (Jun-Seong et al., 2025) | 5 | 37.84 |
| | | 10 | 40.22 |
| | | 20 | 43.59 |
| | | 40 | 45.70 |
| | | 50 | 44.93 |
| | Q-Render (ours) | 5 | 49.98 |
| | | 10 | 50.75 |
| | | 20 | 50.65 |
| | | 40 | **50.85** |
| | | 50 | 50.28 |

Figure 6: Comparison of (a) FPS and (b) mIoU by varying the number of Gaussians participating on rendering. Our Q-Render allows real-time rendering while preserving the mIoU performance. For the fair speed comparison, while using our implementation with the same rasterized Gaussians and 512-D Gaussian features, we only change the sampling strategies among the rasterized Gaussians: V-Render (Mildenhall et al., 2021; Kerbl et al., 2023) uses all intersecting Gaussians, top-$K$ (Jun-Seong et al., 2025) sorts and selects $K$ Gaussians, and ours collect $K$ Quantile Gaussians.

## 5.2 LeRF-OVS dataset.

We compare GS-Mink with previous baselines on LeRF-OVS dataset (Kerr et al., 2023; Qin et al., 2024) where the scenes has sampled for the open vocabulary semantic segmentation task such 'ramen', 'teatime', 'kitchen', *etc*. we maintain to use the same ground truth masks and corresponding text captions for the fair comparison. We use GS-Mink as our baselines and train the model with this dataset. As demonstrated in Table 3, our method outperforms recent studies while having 6-D compressed Gaussian embeddings and 512-D Gaussian embeddings as the same dimensionality as CLIP's image embeddings. For visualization, we also provide our qualitative results in Figure 5.

## 5.3 Control experiments

**Feature renderer.** We conduct ablation study by replacing our Q-Render of our GS-Net with other feature rendering algorithms, such as volume render (Kerbl et al., 2023), and the top-$K$ render (Jun-Seong et al., 2025). The comparison results are described in Figure 6 in terms of rendering speed and accuracy in open-vocabulary semantic segmentation.

In general, Q-Render achieves superior or comparable performance compared to other feature rendering algorithms. In terms of rendering speed, our Q-Render demonstrate up to $1.5\times$ faster speed in comparison with volume rendering (V-Render). Moreover, top-$K$ rendering shows remarkable speed drops as $K$ increases. Such a phenomenon is related to our complexity analysis in Table 1.

For the detailed ablation study about $K$, the number of quantile Gaussians, we achieve the best performance when setting $K = 40$. Nonetheless, the performance becomes converged as $K \geq 10$, and reaches $\sim 50$mIoU. Furthermore, Table 1 shows that top-$K$ render has relatively big performance drop when setting smaller $K$. We deduce that this is because of the discrepancy in the transmittance profile as we visualized in Figure 3.

In another analysis, our Q-render with $K = 40$ outperforms the performance of the volume render. Theoretically speaking, our Q-render is designed to approximate the original transmittance profile by the volume render as stated in section 4.2. Though we do not have concrete experimental supports, we guess that this is related to the potential noise in the optimized 3D Gaussians $\mathcal{G}$ which have some difficulties in representing the precise geometry information due to the limited training images (Zhu et al., 2024) or the 3D Gaussian representation itself (Huang et al., 2024).

**Grid size.** Throughout the paper, we used a grid size of 10.0cm for both GS-Mink and GS-PTv3. We also conducted ablation studies by varying the grid size to 10.0, 5.0, 2.0, 1.0, 0.50, and 0.25 cm. The results of open vocabulary semantic segmentation experiments with different grid sizes are

Table 4: Ablation study for the grid size. Note that the scene scale is aligned metric-unit as described in section A of the appendix. So the grid size is determined as below.

| 3D neural network | Feature renderer | Grid Size (cm) | 19 classes | | 15 classes | | 10 classes | |
|---|---|---|---|---|---|---|---|---|
| | | | mIoU (↑) | mAcc (↑) | mIoU (↑) | mAcc (↑) | mIoU (↑) | mAcc (↑) |
| GS-Mink | Q-Render (K=40) | 10.0 | 47.07 | 58.46 | 49.38 | 62.85 | 59.79 | 75.33 |
| | | 5.0 | **50.39** | **62.14** | **53.14** | **66.39** | **63.37** | 78.30 |
| | | 2.0 | 50.28 | 61.58 | 53.00 | 65.98 | 63.29 | **78.34** |
| | | 1.0 | 45.00 | 55.41 | 48.27 | 60.99 | 60.29 | 75.45 |
| | | 0.5 | 41.12 | 49.72 | 44.40 | 55.32 | 57.05 | 71.46 |
| | | 0.25 | 34.36 | 42.42 | 37.34 | 47.47 | 49.03 | 61.78 |
| GS-PTv3 | | 10.0 | 43.71 | 55.96 | 49.94 | 63.09 | 59.60 | 74.79 |
| | | 5.0 | 48.64 | 59.42 | 51.35 | 64.83 | 62.57 | 78.18 |
| | | 2.0 | 48.99 | 60.36 | 52.39 | 66.05 | 62.57 | 77.70 |

Table 5: Rendering speed on ScanNet scene0006_00 (frame 0). Note that 512[†] is implemented by for-loop iterations, leveraging the original baseline code to render a 512 dimension feature map.

| | LangSplat | | | OpenGaussian | | | GS-Mink (ours) | | |
|---|---|---|---|---|---|---|---|---|---|
| Feature dim. | 3 | 6 | 512[†] | 3 | 6 | 512[†] | 3 | 6 | 512 |
| FPS (↑) | 112.12 | - | 0.65 | - | 71.13 | 0.83 | 172.52 | **80.98** | **28.42** (×43.7↑) |

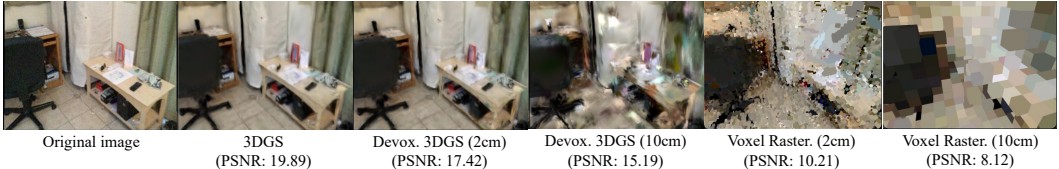

| Original image | 3DGS (PSNR: 19.89) | Devox. 3DGS (2cm) (PSNR: 17.42) | Devox. 3DGS (10cm) (PSNR: 15.19) | Voxel Raster. (2cm) (PSNR: 10.21) | Voxel Raster. (10cm) (PSNR: 8.12) |

Figure 7: Information loss after voxelization. It shows that '*a rendered image from de-voxelized Gaussians*' achieved higher fidelity compared to '*a rendered image directly from sparse voxels*' (w/o de-voxelization).

presented in Table 4. We observed that reducing the grid size to 5.0cm shows the best performance. While the performance started to dramatically drop after setting grid size smaller than 1.0 cm. We deduce this phenomenon from the model architecture and voxelization strategy. While we follow the Gaussian voxelization scheme by (Chen et al., 2024), it only sample one voxel per Gaussian though each Gaussian has volumetric shapes. Moreover, the 3D neural network baselines have some receptive field extent. Once we reduce the grid size smaller than specific numbers, some voxels may not aggregated together due to the receptive field limitations. Based on this analysis, we believe that exploring model designs or voxelization type can become potential future research directions for the further improvements.

**Inference time.** In Table 5, we evaluate the inference speed of recent methods and our GS-Mink on scene0000_00 from ScanNet. For a fair comparison, we measure frames per second (FPS) using the same feature dimension settings as in LangSplat (3 dim) and OpenGaussian (6 dim), and utilize the full Gaussian scene. Our method achieves the highest rendering speed among the evaluated approaches. We modify the official implementation by (Qin et al., 2024) and (Wu et al., 2024b) to render 512-D feature maps by for-loop iterations (we denoted these implementations as 512[†] in Table 5). It turns out that our Q-Render achieves upto ∼**43.7×** speed gains when rendering 512-D feature maps.

**Information loss after voxelization.** We visualize the voxelization results in Figure 7. The voxelization itself potentially brings information loss, and changes the original 3D Gaussian parameters, such as opacity, spherical harmonics, *etc*. To estimate the amount of information loss, we conduct an experiment that renders images from (1) original Gaussians, or (2) voxelized & de-voxelized Gaussians. Figure 7 shows that PSNR drops from 19.89 to 15.19 when $\delta$=0.10. Due to these reasons, we de-voxelize the estimated voxel features from the 3D neural network into Gaussian features. Then, we proceed with our Q-rendering. We used (Sun et al., 2025) to directly render images from sparse voxels without de-voxelization steps.

Table 6: Performance change on the number of $K$. We use a model trained with $K = 40$ and use different numbers of $K$ during inference.

| $K$ | 5 | 10 | 20 | 40 | 50 |
|---|---|---|---|---|---|
| mIoU ($\uparrow$) | 39.16 | 42.18 | 44.94 | 45.81 | 45.71 |
| mAcc ($\uparrow$) | 48.43 | 53.94 | 56.14 | 56.87 | 56.94 |

Table 7: Comparison of GS-Net performance across different input 3D-GS models. GS-Net v2 utilizes 3D Gaussians pre-trained with an additional depth loss as input.

| Method | Training 3D-GS | 19 classes | | 15 classes | | 10 classes | |
|---|---|---|---|---|---|---|---|
| | | mIoU ($\uparrow$) | mAcc ($\uparrow$) | mIoU ($\uparrow$) | mAcc ($\uparrow$) | mIoU ($\uparrow$) | mAcc ($\uparrow$) |
| GS-Net v1 | Rendering | 28.42 | 38.85 | 31.02 | 44.02 | 42.58 | 57.92 |
| **GS-Net v2 (Ours)** | Rendering + Depth | **50.75** | **62.00** | **53.54** | **66.39** | **64.94** | **79.34** |

## 6 CONCLUSION

The core contribution of this work is Quantile Rendering (Q-Render), a transmittance-aware strategy that resolves the computational bottleneck of embedding high-dimensional features in 3D Gaussian Splatting. Unlike conventional rendering that densely accumulates all intersections, Q-Render adaptively samples only the influential 'quantile' Gaussians that dominate a ray's transmittance profile. This sparse sampling drastically reduces computational overhead, enabling efficient rendering of full 512-dimensional feature maps with up to a 43.7x speedup. Consequently, Q-Render achieves state-of-the-art performance on open-vocabulary segmentation benchmarks, establishing it as a scalable bridge between 2D foundation models and 3D representations.

**Limitations and Future Work.** Although Q-Render has shown efficient approximation of volumetric rendering, there still exists several limitations and failure cases.

**Limitation 1: Dynamic $K$ Selection.** Theoretically, the optimal number of samples $K$ should vary depending on the distribution of transmittance along a ray. However, in this work, we use a fixed $K$ across all experiments for simplicity. To investigate the sensitivity of our model to $K$, we conducted an analysis where we trained the model with $K = 40$ and performed inference with varying $K \in \{5, 10, 20, 40, 50\}$. As shown in Table 6, the performance significantly drops when the inference $K$ differs from the training configuration. This underscores the necessity of an adaptive $K$ selection strategy. Although we explored two adaptive strategies in Appendix E.1, they incur high computational costs. Thus, we leave the development of efficient, adaptive sampling strategies for future work.

**Limitation 2: Dependence on 3D-GS.** Our current framework assumes that input 3D Gaussians are obtained through per-scene optimization, which inherently limits practical scalability. However, emerging generalizable 3D-GS approaches that eliminate the need for per-scene optimization, such as DepthSplat (Xu et al., 2025), WorldMirror (Liu et al., 2025), and DepthAnything3 (Li et al., 2024) offer a promising path to resolve this issue. Furthermore, we observe that the quality of the input 3D Gaussians significantly impacts downstream performance. As shown in Table 7, GS-Net v1, which utilizes the original 3D-GS optimized without depth supervision, yields suboptimal results. Conversely, GS-Net v2 takes as input a more recent 3D-GS implementation trained with additional depth loss. This improvement leads to substantial performance gains, highlighting the critical importance of geometric accuracy in the input representation. We belive advancements in 3D-GS will also involve the improvement of GS-Net.

**Limitation 3: Dependence on 3D Network Architecture.** As reported in Appendix E.3, we observe that performance highly depends on the choice of backbone network. In particular, Minkowski-iNet (Choy et al., 2019) and PTv3 (Wu et al., 2024a) exhibit strong sensitivity to the voxel grid resolution, as shown in Table 4. This suggests that developing a more efficient and Gaussian-aware architecture could further improve the performance of our GS-Net framework. Potential directions include introducing Gaussian-friendly operators, reducing or eliminating voxelization, and designing modules that are robust to noise in Gaussian parameters.

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

APPENDIX

## A    RESOLVING UP-TO-SCALE WITH MONOCULAR DEPTH

The scale of a 3D scene is a critical factor in 3D scene understanding. However, when camera poses are estimated solely from multi-view images, recovering the absolute scene scale—referred to as the up-to-scale problem—becomes challenging. For instance, COLMAP Schönberger & Frahm (2016); Schönberger et al. (2016), a widely used tool for extracting extrinsic and intrinsic camera parameters in neural rendering pipelines, does not inherently address this issue. Since our method relies on optimized Gaussian parameters $\Theta$, the up-to-scale problem naturally arises when handling diverse 3D scenes captured purely from images.

To overcome this issue, we employ DepthAnythingV2 Yang et al. (2025), an off-the-shelf monocular depth estimation method trained to produce metric-scale depth maps from single images. Similar to the depth alignment process in Kerbl et al. (2024); Chung et al. (2024), we fix the original Gaussian parameters $\Theta$ and optimize the global scene scale $a \in \mathbb{R}$ to approximate the metric scale:

$$\arg \min_a \sum_{i=1}^{N_{\mathcal{I}}} |\text{invD}_i^{\text{mono}} - a \cdot \text{invD}_i^{\text{rendered}}|_1, \tag{3}$$

where $N_{\mathcal{I}}$ is the number of known images, $\text{invD}_i^{\text{mono}}$ is $i$-th inverse depth map from DepthAnythingV2 Yang et al. (2025), and $\text{invD}_i^{\text{rendered}}$ is the corresponding rendered inverse depth map using 3DGS. After obtaining the scene scale $a$, we update the Gaussian parameter $\Theta$ as $\tilde{\Theta} = \{\tilde{\theta}_i\}_{i=1}^N = \{\frac{1}{a} \cdot \boldsymbol{\mu}_i, \frac{1}{a} \cdot \mathbf{s}_i, \mathbf{r}_i, \alpha_i, \mathbf{sph}_i\}_{i=1}^N$. Note that only the mean and the scale attributes of each Gaussian are changed. We provide an example result in Figure 8.

Unlike the original 3DGS Kerbl et al. (2023), which recently introduced a scale-alignment method, we avoid pruning initial COLMAP points that could affect the distribution of our optimized Gaussian parameters $\Theta$. This ensures a fair comparison with recent studies such as Gaga Lyu et al. (2024). Furthermore, we use a metric depth estimator, *'depth-anything-v2-metric-hypersim-vitl.pth'* in the official repository, whereas 3DGS employs an inverse depth estimator, *'depth-anything-v2-vitl.pth'*. Instead of normalizing the estimated depth maps, we directly take their reciprocal to obtain inverse depth maps $D^{\text{mono}}$.

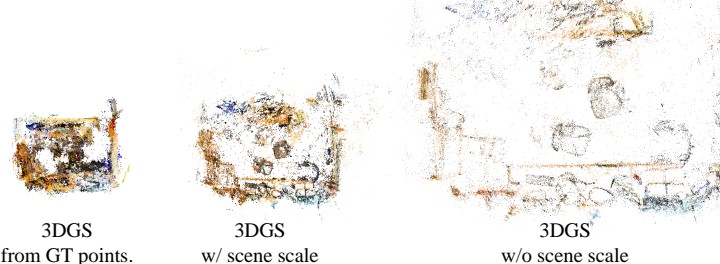

|  |  |  |
|---|---|---|
| 3DGS | 3DGS | 3DGS |
| from GT points. | w/ scene scale | w/o scene scale |

Figure 8: Visualization of 3D Gaussians with different conditions: (left) 3D Gaussians that are trained from ground truth pointcloud, (middle) 3D Gaussians that are trained from COLMAP points, then are applied with scene scale $a$, (right) 3D Gaussians that are trained from COLMAP points without considering scene scale $a$.

## B    DATA PREPROCESSING

### B.1    3D GAUSSIAN SPLATTING OPTIMIZATION

We use the original implementation with the proposed hyperparameters, such as training iterations and upsampling frequency, for pre-optimizing 3D-GS. For the initial points, we follow the initialization strategies used by OpenGaussian Wu et al. (2024b) and Dr.Splat Jun-Seong et al. (2025), utilizing the point clouds provided by the original ScanNetv2 dataset instead of using

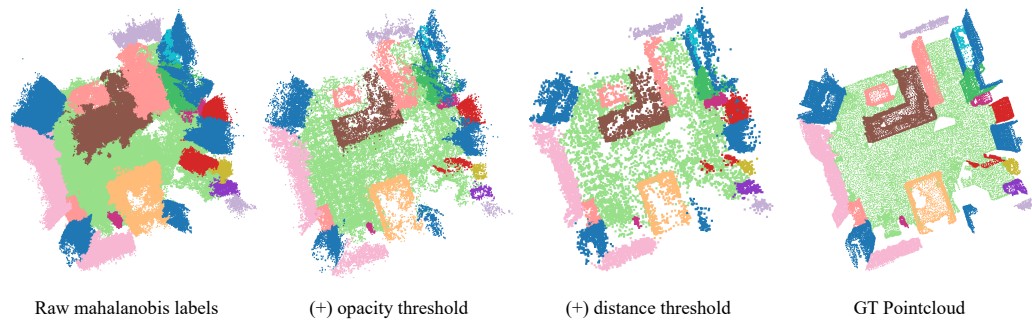

| Raw mahalanobis labels | (+) opacity threshold | (+) distance threshold | GT Pointcloud |

Figure 9: Generated Gaussian Splatting labels from point cloud labels. Applying both opacity and distance thresholding allows much clearer label generation for Gaussian Splatting.

COLMAP Schönberger & Frahm (2016); Schönberger et al. (2016). Note that we maintain the densification and pruning processes of 3D-GS to preserve its capacity.

### B.2 Pseudo ground truth acquisition

Since we enable the densification and pruning processes of 3D-GS, the number and coordinates of each Gaussian differ from the original point clouds. To evaluate the predicted Gaussians, we need to create Gaussian-specific labels derived from the original point labels. As shown in Figure 9, for each Gaussian, we first explore the $K$ nearest neighbor points and assign the label based on the most frequent label among these neighbors. However, we observed that due to the large number of Gaussians, many Gaussians contribute minimally to the scene (i.e., their opacity is below 0.1), and we filtered out these less significant Gaussians. Despite this, we found that some isolated Gaussians (i.e., "floaters") still received labels, resulting in noisy ground truth (GT) labels. To address this, we ignored labels if all the nearest points had a Mahalanobis distance above 0.1, effectively removing such noisy assignments. As a result, we were able to obtain clearer and more consistent Gaussian labels.

## C Theoretical Analysis of Quantile Rendering

Based on the definitions provided in Section 4.2 and Algorithm 1 of the manuscript, we provide the theoretical analysis of Quantile Rendering (Q-Render) as an approximation of the volume rendering in 3D-GS (Kerbl et al., 2023) using Riemann sums.

### C.1 Preliminaries: Riemann Sums and the Right Rule

Let $f : [a, b] \to \mathbb{R}$ be a bounded function defined on a closed interval $[a, b]$. We define a partition $P$ of the interval into $N$ sub-intervals:

$$a = u_0 < u_1 < u_2 < \cdots < u_N = b$$

The width of the $k$-th sub-interval $[u_{k-1}, u_k]$ is denoted by $\Delta u_k = u_k - u_{k-1}$. A **Riemann Sum** approximates the definite integral of $f$ over $[a, b]$ by summing the areas of rectangles:

$$S = \sum_{k=1}^{N} f(u_k^*) \Delta u_k \tag{4}$$

where $u_k^*$ is a sample point chosen from the sub-interval $[u_{k-1}, u_k]$.

The **Right Riemann Sum** specifically chooses the right endpoint of each sub-interval as the sample point:

$$u_k^* = u_k. \tag{5}$$

Thus, the approximation becomes:

$$S_{right} = \sum_{k=1}^{N} f(u_k) \Delta u_k \tag{6}$$

When the partition is uniform, $\Delta x = (b-a)/N$, due to the 'Right endpoint approximation theorem', the Right Rule converges linearly:

$$\left| \int_a^b f(x)\,dx - S_{\text{right}} \right| \leq \frac{M(b-a)^2}{2N},$$

where $M$ is the maximum value of the absolute value of $\frac{\partial f(x)}{\partial x}$ over the interval.

## C.2 VOLUME RENDERING AS AN INTEGRAL

We observe that the discrete volume rendering process is fundamentally equivalent to a Riemann sum approximation of a continuous path integral. Theoretically, volume rendering assumes that the accumulated transmittance $T$ monotonically decreases from 1 to 0 along the ray, creating a bounded integration domain. Different neural rendering approaches can be characterized by how they partition this domain:

- **NeRF (Mildenhall et al., 2021):** The integral is approximated in the *spatial domain* ($t$). Query points are sampled uniformly or hierarchically (coarse-to-fine) along the ray. The Riemann partition is defined by the distance between adjacent samples ($\delta_i = t_{i+1} - t_i$), where the opacity $\alpha_i$ is derived from the learned volumetric density $\sigma$.
- **3D-GS (Kerbl et al., 2023):** The integral is evaluated using a set of discrete primitives sorted by depth. Here, the rasterized Gaussians act as the representative samples for intervals along the ray. The rendering equation accumulates the alpha-blended contribution of each intersecting Gaussian, effectively performing a Riemann summation where the step size and weight are determined by the Gaussian's covariance and opacity.
- **Quantile Rendering (Ours):** Unlike the spatial sampling in NeRF or the dense accumulation in 3D-GS, our method shifts the Riemann partition to the *transmittance domain* ($T$). By sampling intervals of equal probability in transmittance space, we ensure that the summation is performed only over the most significant contributors, offering a more efficient approximation of the underlying integral.

## C.3 VOLUME RENDERING AS A CONTINUOUS INTEGRAL

Consider a camera ray parameterized by distance $t \in [0, \infty)$. Let $\sigma(t)$ denote the volume density and $c(t)$ the feature (or color) emitted at position $t$.

The transmittance is defined as

$$T(t) = \exp\left( -\int_0^t \sigma(s)\,ds \right).$$

Its derivative satisfies

$$\frac{dT}{dt} = -\sigma(t)T(t) \quad \Longrightarrow \quad -\frac{dT}{dt}\,dt = \sigma(t)T(t)\,dt.$$

The continuous volume rendering integral is

$$C_{\text{vol}} = \int_0^\infty c(t)\,\sigma(t)T(t)\,dt. \tag{7}$$

The Quantile render operates by analyzing the change in Transmittance $T$ rather than spatial distance $t$. We can reformulate the volume rendering integral Eq. 7 by performing a change of variables from spatial distance $t$ to transmittance $u = T(t)$. Following (Mildenhall et al., 2021; Kerbl et al., 2023), $u$ and $t$ are noted as:

- When $t = 0$, $u = 1$ (Ray starts with full transmittance starting from 1).

- When $t \to \infty$, $u \to 0$ (Transmittance converges to 0).

Since $T$ is strictly decreasing from 1 to 0, the substitution

$$u = T(t), \qquad du = -\sigma(t)T(t)\,dt$$

transforms Eq. 7 into

$$C_{\text{vol}} = \int_{u=1}^{0} c(u)\,(-du) = \int_{0}^{1} c(u)\,du.$$

Thus, volume rendering is exactly the integral of a function $c(u)$ over the transmittance interval $[0, 1]$:

$$C_{\text{vol}} = \int_{0}^{1} c(u)\,du. \tag{8}$$

### C.4   QUANTILE RENDERING AS A RIGHT RIEMANN SUM

As described in Section 4.2 of the manuscript, the Quantile Rendering algorithm partitions the Transmittance $T \in [0, 1]$ into $K + 1$ evenly distributed segments. The algorithm selects a specific 'Quantile Gaussian' whenever the accumulated transmittance drops.

In detail, the Quantile Rendering partitions the transmittance domain $[0, 1]$ into

$$0 = u_{K+1} < u_K < \cdots < u_1 < u_0 = 1.$$

For each interval $[u_{k-1}, u_k]$, Q-render selects the Gaussian whose transmittance crossing corresponds to the right endpoint $u_k$. Thus the algorithm evaluates $c(u)$ at $u_k$. Therefore, Q-render implements exactly the Right Riemann Sum for Eq. 8:

$$C_Q^{\text{right}} = \sum_{k=1}^{K+1} c(u_k)\,\Delta u. \tag{9}$$

This replaces the dense per-Gaussian accumulation with a sparse, quantile-driven sampling strategy.

### C.5   APPROXIMATION ERROR OF QUANTILE RENDERING

Following (Mildenhall et al., 2021; Kerbl et al., 2023), the integrand $c(u)$ is differentiable and satisfies

$$|c'(u)| \le M \quad \forall u \in [0, 1].$$

Applying the classical Right Rule error bound with $a = 0$, $b = 1$, and $N = K + 1$ gives:

$$\left| C_{\text{vol}} - C_Q^{\text{right}} \right| \le \frac{M}{2(K+1)} \le \frac{M}{2K}. \tag{10}$$

Thus the quantile approximation converges at rate $O(1/K)$.

### C.6   CONVERGENCE OF Q-RENDER WITH TRANSMITTANCE NORMALIZATION

Q-render includes a final normalization step to correct for residual transmittance:

$$\widetilde{C}_Q = \frac{C_Q^{\text{right}}}{1 - T_Q}, \tag{11}$$

where $T_Q$ is the remaining transmittance after processing all $K$ quantile Gaussians.

Since each quantile removes at least $\Delta u$, we have

$$T_Q \le \Delta u = \frac{1}{K+1}. \tag{12}$$

Hence the normalization factor satisfies

$$\frac{1}{1 - T_Q} \leq \frac{1}{1 - \frac{1}{K+1}} = \frac{K+1}{K}. \tag{13}$$

Combining with eq. 10 yields

$$\left| C_{\text{vol}} - \widetilde{C}_Q \right| \leq \frac{K+1}{K} \cdot \frac{M}{2(K+1)} = \frac{M}{2K}. \tag{14}$$

**Final Theorem.**   Under the assumptions that $c$ is differentiable and $|c'(u)| \leq M$ (Mildenhall et al., 2021; Kerbl et al., 2023), Q-render converges to volume rendering with rate

$$\boxed{\left| C_{\text{vol}} - \widetilde{C}_Q \right| \leq \frac{M}{2K}} \tag{15}$$

and the approximation error vanishes linearly as $K \rightarrow \infty$.

# D   IMPLEMENTATION DETAILS

For optimization, we use the default optimizers provided by MinkUNet (Adam) and PTv3 (AdamW). The learning rate is adjusted using PyTorch's ReduceLROnPlateau scheduler, reducing by a factor of 10 when a plateau is detected. Training is conducted with a batch size of 4 across 8 A100-80GB GPUs while computing rendering loss from 4 randomly chosen training viewpoints. Both MinkUNet and PTv3 follow their default configurations.

# E   ADDITIONAL EXPERIMENTS

## E.1   ADAPTIVE GAUSSIAN SAMPLING STRATEGY

We additionally implement two adaptive variants of Q-Render, namely `Learned-K` and `Stratified-K`, to assess whether adaptive selection of $K$ can effectively mitigate the sensitivity to this hyperparameter.

**Learned-K.**   In this variant, we train the model with three candidate values for $K$, specifically $\{10, 20, 40\}$. Inspired by the mIoU-prediction head in SAM2, we introduce an additional *similarity-prediction head* that estimates the expected similarity between the predicted features and the target features for each candidate $K$. This head takes the transmittance profile as input and outputs the predicted similarity scores. Therefore, similar to SAM2, we use an additional loss that computes the difference between predicted similarity and the actual similarity.

$$\mathcal{L}_{sim}(\tilde{\mathbf{f}}^Q, \mathbf{f}_i^{\text{CLIP}}, \{T_n\}_{n=1}^N) = || \text{Sim-Head}_\Theta(\{T_n\}_{n=1}^N) - \text{sim}(\tilde{\mathbf{f}}^Q, \mathbf{f}_i^{\text{CLIP}})||_2^2, \tag{16}$$

where $\{T_n\}_{n=1}^N$ are transmittance values along the ray, $\tilde{\mathbf{f}}^Q$ is a rendered feature via Q-Render, and $\mathbf{f}_i^{\text{CLIP}}$ is the CLIP feature used in training.

In addition, we apply the original loss for three $K$s during training to optimize features for all $K$s. During inference, the model evaluates these similarity scores and dynamically selects the value of $K$ that yields the highest expected similarity. This allows the model to adaptively determine $K$ per input without relying on a manually fixed value.

**Stratified-sampling.**   In the second variant, we compute the mean and standard deviation of the transmittance values along the depth dimension. Rather than uniformly partitioning the transmittance for sampling, we instead draw samples based on the *z-score* under a Gaussian distribution parameterized by the estimated mean and standard deviation. This enables denser sampling at depths where objects are more likely to contribute, analogous to the stratified sampling strategy used in

Table 8: Open vocabulary 3D semantic segmentation performances in the ScanNet dataset.

| Method | FPS | 19 classes | | 15 classes | | 10 classes | |
|--------|-----|-----------|-----------|-----------|-----------|-----------|-----------|
| | | mIoU (↑) | mAcc (↑) | mIoU (↑) | mAcc (↑) | mIoU (↑) | mAcc (↑) |
| Learned-K | 14.31 | 40.14 | 49.16 | 43.85 | 55.60 | 55.84 | 68.53 |
| Stratified-K | 15.14 | **41.30** | 49.52 | **44.67** | **55.78** | 56.66 | 70.10 |
| Q-Render (ours) | **32.17** | 41.12 | **49.72** | 44.40 | 55.32 | **57.05** | **71.46** |

Table 9: Comparison between Dr.Splat and GS-Mink on MipNeRF360 benchmark.

| Scene | Dr.Splat | | GS-Mink | |
|-------|----------|----------|---------|----------|
| | mIoU | mAcc | mIoU | mAcc |
| bicycle | 0.2112 | 0.3012 | **0.2236** | **0.3165** |
| garden | 0.5543 | 0.6114 | **0.6721** | **0.7813** |
| treehill | **0.2122** | **0.2713** | 0.2063 | 0.2585 |
| Avg. | 0.3359 | 0.3946 | **0.3673** | **0.4521** |

NeRF, and improves robustness under diverse transmittance distributions. Specifically, we uniformly partition the transmittance interval in the $z$-score space and then map the samples back to the original transmittance domain. Consequently, unlike the original Q-Render–which uniformly partitions the transmittance values–this variant adapts the sampling locations according to the underlying transmittance distribution, encouraging the model to draw more Gaussians from regions where they are more densely concentrated.

**Results.** We compare these two adaptive variants against the original Q-Render in terms of segmentation performance and rendering FPS. Due to the large memory footprint of the `Learned-K` model, we use a voxel size of 0.5 for all experiments. For a fair comparison, we set $K = 40$ for both the `stratified-K` variant and the original Q-Render. As shown in Table 8, the `Learned-K` model slightly underperforms the other strategies, likely because the model must additionally learn the similarity-prediction head, which makes it harder to focus on improving feature quality. We also observe that the `stratified-K` variant performs on par with the original Q-Render. However, the most critical observation is that both adaptive variants achieve nearly half the FPS of Q-Render. This slowdown arises because they require two passes along each ray: one to estimate the transmittance statistics and another to perform rendering. Despite their increased computational cost, we do not observe any clear performance improvement over Q-Render. Therefore, we adopt Q-Render as our rendering algorithm of choice, as it provides an efficient and effective approximation of V-Render.

### E.2 OPEN VOCABULARY SEGMENTATION ON OUTDOOR SCENES

Since our evaluation benchmarks are all indoor datasets, we also demonstrate the effectiveness of our proposed method on outdoor benchmarks. However, to the best of our knowledge, there are currently no benchmarks that jointly cover multi-view and outdoor scenes for open-vocabulary segmentation (OVS). To enable comparison in outdoor settings, we manually annotated 3 scenes from MipNeRF360 (Barron et al., 2022) outdoor scenes. Specifically, we use SAM2 to obtain an initial mask corresponding to the queried object and then manually refine the segmentation by providing additional positive and negative point prompts. This process allows for reliable ground-truth masks for evaluating outdoor multi-view OVS performance across all baselines.

We compare our framework with the state-of-the-art method, Dr.Splat (Jun-Seong et al., 2025), using the manually annotated MipNeRF360 benchmark. As reported in Table 9, Q-Render consistently outperforms Dr.Splat, corroborating our main findings and demonstrating robust generalization to outdoor scenarios. Qualitative comparisons are visualized in Figure 10. As observed in the first column, our method achieves clear separation between the target object and its surroundings. However, we acknowledge that VRAM constraints necessitated the use of larger voxel sizes. We leave the development of more memory-efficient network architectures for 3D-GS to future work.

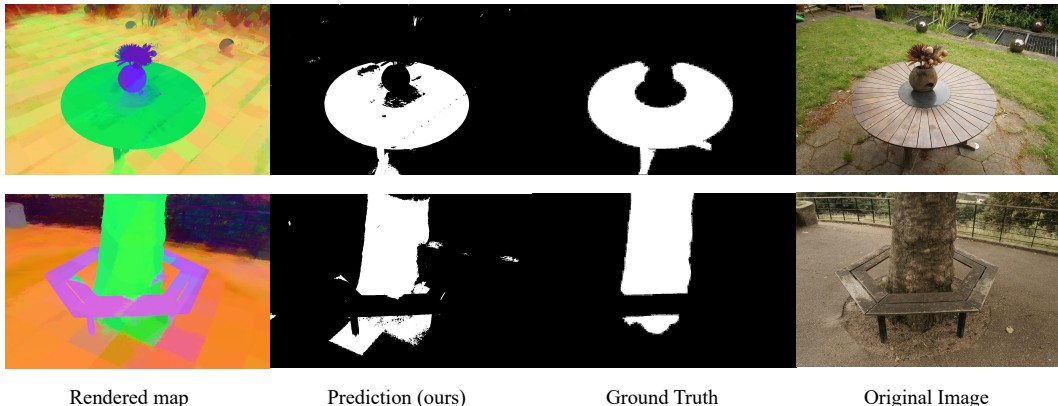

| Rendered map | Prediction (ours) | Ground Truth | Original Image |

Figure 10: Our qualitative results in the MipNeRF360 dataset.

Table 10: Comparison of various 3D network backbone to GS-Net.

| 3D arch. | Rendering | 19 classes | | 15 classes | | 10 classes | |
|---|---|---|---|---|---|---|---|
| | | mIoU (↑) | mAcc (↑) | mIoU (↑) | mAcc (↑) | mIoU (↑) | mAcc (↑) |
| PointNet++ (Qi et al., 2017) | Q-render | 39.42 | 50.62 | 42.88 | 56.66 | 52.21 | 69.14 |
| PointNeXT (Qian et al., 2022) | | 37.89 | 45.80 | 41.15 | 52.09 | 51.57 | 63.83 |
| PTv3 (Wu et al., 2024a) | | 48.99 | 60.36 | 52.39 | 66.05 | 64.95 | 79.34 |
| MinkUNet (Choy et al., 2019) | | 50.75 | 62.00 | 53.54 | 66.39 | 64.95 | 79.34 |

### E.3 MORE NETWORK ARCHITECTURE

To evaluate the generalizability of our approach across different backbones, we conducted additional experiments using two representative point-based architectures: PointNet++ (Qi et al., 2017) and PointNeXT (Qian et al., 2022). PointNet++ utilizes hierarchical feature learning via local neighborhood grouping to capture fine-grained geometry, while PointNeXT advances this design with residual connections and improved scaling strategies. However, as reported in Table 10, both point-based models exhibit inferior performance compared to voxel-based baselines (MinkUNet and PTv3). We conjecture that this disparity stems from the handling of point density. Since 3D Gaussians are often densely clustered around object surfaces, the k-nearest neighbor (k-NN) search used in point-based networks tends to limit the metric receptive field, thereby hindering the aggregation of broader contextual information. In contrast, voxel-based networks are inherently more robust to such density variations, resulting in superior performance.

### E.4 THOROUGH COMPARISON WITH OPENGAUSSIAN

To better understand which components of our method contribute most to the performance improvement, we conduct an ablation by modifying each part of the pipeline individually and comparing the results against OpenGaussian. Specifically, we introduce two hybrid models: one that uses the GS-Net feature extractor while retaining the original rendering module of OpenGaussian, and another that keeps the OpenGaussian feature extractor but replaces the renderer with Q-Render. As shown in Table 11, incorporating Q-Render yields a consistent performance gain for both OpenGaussian and GS-Net. Moreover, replacing the OpenGaussian codebooks with GS-Net features leads to a substantial improvement. We conjecture that preserving high-dimensional features is a key factor, as OpenGaussian encodes only 6-dimensional attributes, whereas GS-Net leverages 512-dimensional CLIP features that retain richer semantic information.

### E.5 SENSITIVITY UNDER GAUSSIAN PERTURBATION

We also investigate the sensitivity of GS-Net to perturbations applied to the input 3D-GS representation. Specifically, on top of the predicted opacity, we inject Gaussian noise with varying scales into the Gaussian opacity values and evaluate the resulting segmentation performance. To avoid extreme

Table 11: Comparison of V-Render and Q-Render under different feature extractors.

| Feature Extractor | Rendering | 19 classes | | 15 classes | | 10 classes | |
|---|---|---|---|---|---|---|---|
| | | mIoU (↑) | mAcc (↑) | mIoU (↑) | mAcc (↑) | mIoU (↑) | mAcc (↑) |
| OpenGaussian | V-Render | 22.60 | 34.41 | 24.21 | 37.58 | 34.74 | 51.58 |
| | Q-Render | 23.10 | 35.55 | 26.18 | 38.12 | 36.13 | 56.12 |
| GS-Net (Mink) | V-Render | 49.02 | 58.75 | 50.41 | 63.65 | 61.04 | 76.21 |
| | Q-Render | 50.75 | 62.00 | 53.54 | 66.39 | 64.94 | 79.34 |

Table 12: Robustness of Q-Render to Gaussian noise added to opacity.

| Noise Scale | 19 classes | |
|---|---|---|
| | mIoU (↑) | mAcc (↑) |
| 0 | 50.75 | 62.00 |
| 0.25 | 50.13 | 60.94 |
| 0.5 | 50.18 | 60.88 |
| 1 | 47.13 | 57.10 |
| 2 | 38.13 | 44.11 |
| 4 | 16.12 | 32.19 |

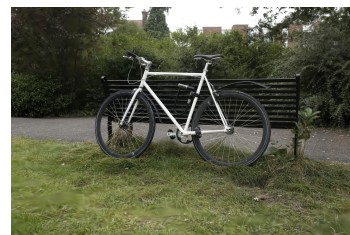 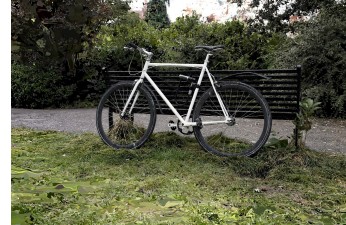 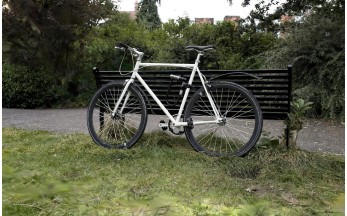

Volumetric rendering
59.429 FPS (1063 x1600 image)

Quantile rendering (K=1)
70.065 FPS (1063 x1600 image)

Quantile rendering (K=10)
66.087 FPS (1063 x1600 image)

Figure 11: Comparison between V-Render and Q-Render on RGB reconstruction using a pre-trained 3D-GS model without fine-tuning. Q-Render shows only minor PSNR degradation, demonstrating that it serves as an efficient and accurate approximation of V-Render in the RGB domain.

cases where the opacity becomes negative values, we apply the noise before the sigmoid activation is applied. As shown in Table 12, GS-Net remains highly robust under mild noise levels: performance degradation at noise scales of 0.25 and 0.5 is negligible, demonstrating that Q-Render retains sufficient stability even when the opacity field is moderately corrupted. However, as the noise scale increases beyond 1.0, we observe a marginal drop in both mIoU and mAcc, and the performance eventually collapses at extreme noise levels such as 4.0. These results indicate that while Q-Render is resilient to realistic perturbations in the opacity field, strong distortions that heavily corrupt the underlying geometry inevitably degrade performance. Overall, this experiment suggests that our pipeline maintains robust performance under practical levels of noise in 3D-GS inputs.

## E.6 RENDERING RGB

Since our Q-Render is not restricted to CLIP features, it could also be applied to RGB image rendering. Using this setup, we quantify how much information is lost when replacing V-Render with Q-Render. To isolate the behavior of the renderer itself, we directly apply Q-Render to pre-trained 3D-GS models without any fine-tuning. As shown in Fig. 11, Q-Render exhibits only a very slight drop in PSNR compared to V-Render, indicating that it provides an effective approximation of the original renderer even in the RGB domain. This suggests that Q-Render is broadly applicable beyond our semantic feature rendering setup. In particular, we believe Q-Render can be seamlessly integrated into various 3D-GS variants—including dynamic or time-varying Gaussians—since these models also assume fixed Gaussian positions at each rendering step.

Table 13: Peak inference memory on scene0000_00 (ScanNet, 100 frames).

| Method | Feature Dim. | Peak Memory (GB) |
|---|---|---|
| LangSplat | 3 | 7.18 |
| OpenGaussian | 6 | 16.13 |
| Dr.Splat | 512 | 61.13 |
| Q-Render ($K=40$) | 512 | 27.18 |

### E.7 MEMORY FOOTPRINT COMPARISON

We also compare the VRAM requirements of each baseline on scene0000_00 from ScanNet (100 frames). As summarized in Table 13, LangSplat and OpenGaussian require little memory due to their low-dimensional feature representations (3D and 6D). However, this compression inevitably introduces information loss, which correlates with their reduced segmentation performance.

Dr.Splat relies on 512D features and stores per-view Gaussian visibility masks, which causes memory usage to grow proportionally with the number of frames and leads to more than 61,GB of peak consumption. Q-Render also uses 512D features, and its cache-free per-ray accumulation avoids this overhead, reducing the peak memory to 27.18,GB at $K=40$ while preserving high-dimensional semantic capacity.

## F PSEUDO-CODE FOR VOXELIZATION, VOXEL FEATURE COMPOSITION, AND DE-VOXELIZATION

To ensure reproducibility, we provide detailed pseudo-code of our implementation. Listing 1 outlines the voxelization pipeline that groups Gaussians into voxels for efficient processing. Listing 2 details the organization of voxel positions and features as inputs for the 3D network. Finally, Listing 3 describes the de-voxelization process, where voxel features are redistributed to predict Gaussian-level features.

```python
from typing import List, Tuple
from typing_extensions import Literal

# Global variable indicating the shape to be sampled
SAMPLE_SHAPE: Literal["volume", "tri-plane", "tri-line", "center"] = "volume"

def voxelization(gaussians, grid_size):
    """
    Convert a set of 3D Gaussians into a set of unique voxels and their associated features.
    This function does the following:
      1. Counts how many voxels will be needed for each Gaussian based on the current
         SAMPLE_SHAPE (e.g., "volume", "tri-plane", etc.).
      2. Generates voxel coordinates and features for all Gaussians.
      3. Removes duplicates, resulting in a unique set of voxel coordinates, their
         corresponding features, and an index map that indicates how each original
         voxel maps to the new unique list.
    Args:
        gaussians : List[dict]
            A list of 3D Gaussian parameters. Each Gaussian is typically a dictionary
            containing at least a "scale" key (and possibly position or other attributes)
            needed by the downstream functions.
        grid_size : float
            A hyperparameter defining the size of each voxel.
    Returns:
        uni_voxels_xyz : List[Tuple[float, float, float]]
            The (x, y, z) coordinates of the unique voxels.
        uni_voxel_feats : List
            The feature vectors corresponding to each unique voxel.
        inverse_indices : List[int]
            A list of indices that maps each original voxel back to its corresponding
            index in the unique voxel list.
        num_voxels_per_g: List[int]
            A list of number of voxels that are sampled from each 3D Gaussian.
    Notes:
        - The function relies on three helper functions which are assumed to be defined:
            1) `count_voxels_per_g(gaussians, grid_size)`
               Determines the number of voxels for each Gaussian given the sampling shape.
            2) `compose_voxel_features(gaussians, num_voxels_per_g)`
               Produces the (x, y, z) coordinates and any features for each voxel.
            3) `return_unique_voxels(voxels_xyz, voxel_feats)`
               Removes duplicate voxel coordinates and returns a list of unique
               coordinates, their combined features, and an inverse index array.
        - If you are implementing this in a parallel computing environment (e.g., CUDA),
          each step may be adapted for batch or GPU-based processing.
    """
    # 1. Determine how many voxels are required for each Gaussian.
    num_voxels_per_g = count_voxels_per_g(gaussians, grid_size)

    # 2. Generate voxel coordinates (xyz) and their associated feature vectors.
    #    These features might include Gaussian-related attributes or other data.
    voxels_xyz, voxel_feats = compose_voxel_features(gaussians, num_voxels_per_g)

    # 3. Deduplicate the generated voxels:
    #    - Get unique voxel coordinates,
    #    - Aggregate/merge features for any duplicates,
    #    - Obtain the inverse index mapping from original voxels to unique.
    uni_voxels_xyz, uni_voxel_feats, inverse_indices = return_unique_voxels(voxels_xyz,
    ↪  voxel_feats)

    return uni_voxels_xyz, uni_voxel_feats, inverse_indices, num_voxels_per_g
```

Listing 1: Our voxelization process for 3D Gaussians.

```python
from torch import Tensor

def compose_voxel_features(gaussians, num_voxels_per_g, grid_size):
    """
    Generate per-voxel feature vectors for a set of 3D Gaussians.
    This function iterates through each Gaussian and computes:
      1. The xyz location of each voxel in the Gaussian.
      2. A Mahalanobis distance-based influence (voxel opacity).
      3. Additional attributes such as RGB.
    Args:
        gaussians : List[dict]
            A list of Gaussian definitions
        num_voxels_per_g : List[int] or torch.Tensor
            Number of voxels associated with each Gaussian (e.g., from
            `count_voxels_per_g` or another precomputation).
        grid_size : float
            The size of each voxel cell, used to help determine voxel coordinates.
    Returns:
        voxel_features : List[torch.Tensor]
            A list of per-voxel feature tensors having [voxel_xyz, voxel_rgb, voxel_opacity].
    Notes:
        The function `get_voxel_location(...)` is assumed to provide the precise xyz
        ↪ coordinate of each voxel. Implementation details are omitted for readibility.
    """

    voxel_features = []
    voxel_xyzs = []

    # Iterate over Gaussians
    for g_idx, gaussian in enumerate(gaussians):
        # Retrieve the number of voxels expected for this Gaussian
        num_voxels = num_voxels_per_g[g_idx]

        # Extract Gaussian parameters for clarity
        mu = gaussian["mean"]  # Center location of a 3D Gaussian
        inv_cov = gaussian["inverse_3d_covariance_matrix"]
        opacity_factor = gaussian["opacity"]
        color_rgb = gaussian["rgb"]

        # Generate features for each voxel in this Gaussian
        for voxel_idx in range(num_voxels):
            # 1. Get the 3D coordinate for the current voxel
            voxel_xyz = get_voxel_location(voxel_idx, gaussian, grid_size)
            voxel_xyzs.append(voxel_xyz)

            # 2. Compute the Mahalanobis distance, Eq. 1 of the main manuscript
            dist = (voxel_xyz - mu).T @ inv_cov @ (voxel_xyz - mu)

            # 3. Compute voxel opacity using a Gaussian attenuation factor
            voxel_opacity = opacity_factor * torch.exp(-0.5 * dist)

            # 4. Concatenate features
            feature_vec = torch.cat([voxel_xyz, color_rgb, voxel_opacity], dim=1)
            voxel_features.append(feature_vec)

    return voxel_xyzs, voxel_features
```

Listing 2: Pseudocode for the compose-voxel-features function.

```python
from typing_extensions import Literal
from torch_scatter import scatter, segment_csr

# Global reduction mode (could also be passed as a parameter)
REDUCE: Literal["mean", "max"] = "max"

def de_voxelization(uni_voxels_pred, inverse_indices, num_voxels_per_g):
    """
    Aggregate network predictions from unique voxels back to full voxel predictions,
    and then further reduce them into per Gaussian predictions.
    This function performs two main steps:
      1. Unique Voxels -> All Voxels: Uses `scatter` to expand the predictions
         from the unique-voxel level back to the original list of all voxels, by
         indexing through `inverse_indices`.
      2. All Voxels -> Gaussians: Uses `segment_csr` to combine per-voxel
         predictions for each Gaussian. This is done by summing the number of
         voxels per Gaussian in `num_voxels_per_g`, which creates offsets used by
         `segment_csr` to group voxel predictions into Gaussian predictions.
    Args:
        uni_voxels_pred : torch.Tensor
            A tensor containing the predicted values (e.g., features or logits) for each
            unique voxel. Its shape could be (N_unique, C), where N_unique is the number
            of unique voxels and C is the dimensionality of the prediction (e.g. channels).
        inverse_indices : torch.Tensor
            A 1D tensor of indices mapping every original voxel to its corresponding
            index in the unique voxel array.
        num_voxels_per_g : torch.Tensor
            A 1D tensor indicating how many voxels belong to each Gaussian.
    Returns:
        gs_pred : torch.Tensor
            A tensor containing the final predictions at the Gaussian level, with shape
            (G, C).
    Notes:
        - If your pipeline needs a different reduction mode (e.g. `"sum"`) or you want
          to pass it in as a parameter, replace `REDUCE` accordingly.
    """

    # 1. Expand predictions from unique voxels to all voxels using `scatter`.
    #    - 'uni_voxels_pred' has shape (N_unique, C).
    #    - 'inverse_indices' has shape (M, ) with M >= N_unique (the total number of voxels).
    #    - 'scatter' will produce a new tensor of shape (M, C), where each entry is
    #       the value from the appropriate unique voxel. The method of combination
    #       (mean, max, etc.) is determined by `REDUCE`.
    voxels_pred = scatter(uni_voxels_pred, index=inverse_indices, reduce=REDUCE)

    # 2. Combine per-voxel predictions into Gaussian-level predictions using `segment_csr`.
    #    - We first compute offsets with cumulative sums of the number of voxels in each
    #      Gaussian.
    #    - The 'segment_csr' function then takes the voxel predictions and segments them
    #       into groups corresponding to each Gaussian, reducing via the same `REDUCE` rule.
    offset = torch.cumsum(num_voxels_per_g, dim=0)
    gs_pred = segment_csr(voxels_pred, ptr=offset, reduce=REDUCE)

    return gs_pred
```

Listing 3: Our de-voxelization process for 3D Gaussians.

