# OpenReview forum: "Quantile Rendering: Efficiently Embedding High-dimensional Feature on 3D Gaussian Splatting"
_ICLR.cc/2026/Conference — Submitted to ICLR 2026_

### Official Review · Reviewer_hrzW · 2025-10-20

**Soundness:** 2
**Presentation:** 2
**Contribution:** 2
**Rating:** 4
**Confidence:** 3

**Summary:**

This paper introduces Quantile Rendering (Q-Render), an efficient feature rendering algorithm for 3D Gaussian Splatting. Q-Render sparsely samples quantile Gaussians that dominate transmittance along each ray, cutting complexity from O(NC) to O(N+KC). Integrated into a Gaussian Splatting Network (GS-Net), it enables scalable training with high-dimensional CLIP features for open-vocabulary 3D segmentation. Experiments on ScanNet and LeRF-OVS show state-of-the-art accuracy and up to 43.7× faster rendering.

**Strengths:**

1. Efficient Rendering Design: Q-Render introduces a principled quantile-based sampling strategy that substantially reduces computation for high-dimensional feature rendering without sacrificing accuracy.
2. Solid Integration & Generality: The method integrates seamlessly into 3D Gaussian Splatting pipelines and generalizes well across neural backbones, bridging 2D foundation models and 3D representations effectively.
3. The paper is well written and clearly organized.

**Weaknesses:**

1. In figure 6, I think the performance improvement of Q-Render is minor and I think the motivation of the authors apply the quantile rendering is not very strong.
2. The feed-forward pipeline for 3d scene understanding has already been applied in early methods, like SIU3R (SIU3R: Simultaneous Scene Understanding and 3D Reconstruction Beyond Feature Alignment) and SegMASt3R (SegMASt3R: Geometry Grounded Segment Matching).
3. No demo has been submitted; therefore, the performance of the proposed method cannot be effectively demonstrated.

**Questions:**

1. What are the number of input views?
2. What is the training time / GPU memory?

---

> ### Author Response · Authors · 2025-11-27
> **Reviewer hrzW**
>
> **[hrzW, W1] Minor performance improvements of Q-Render**
>
>
> We would like to respectfully clarify that the primary motivation of Q-Render is not to enhance feature fidelity, but to substantially accelerate rendering for high-dimensional features. While the mIoU improvements may appear modest, the 1.4× speed-up in rendering is the key contribution of our method. Achieving near real-time performance (≈30 FPS) is crucial for interactive graphics applications, where system responsiveness directly affects usability and overall user experience.
> We kindly ask the reviewer to reconsider this point: **Q-Render is not designed to significantly surpass V-Render in segmentation accuracy, but rather to deliver fast inference while maintaining comparable performance.**
>
> **[hrzW, W2] Comparison with SIU3R and SegMASt3R**
>
> We would like to respectfully clarify that the mentioned methods are not suitable baselines for our ICLR submission due to both timing and task incompatibility.
>
> First, **SegMASt3R** was posted on arXiv on (2025.10.06), which is after the ICLR submission deadline (2025.09.24). Thus, it was unfortunately not possible for us to be aware of or evaluate this work at the time of submission.
>
> Similarly, the code for **SIU3R** was released on 2025.09.19, only five days before the deadline. This provided insufficient time to conduct a fair and comprehensive evaluation on our benchmarks. In addition, SIU3R employs **CLIP** as its text encoder, whereas our evaluation pipeline is built upon **OpenCLIP**. For a fair comparison, SIU3R would need to be retrained entirely with OpenCLIP, which was not feasible within the limited time frame.
>
> We kindly ask the reviewer to reconsider this point and hope for your understanding regarding the practical constraints we faced at submission time.
>
> **[hrzW, W3] Missing demo**
>
> We have updated a video demo in supplementary materials. Please refer to the attached file.
>
> **[hrzW, Q1] [hrzW, Q2] Questions regarding the implementation details**
>
> For ScanNet experiments, we uniformly sampled 100 frames from the full sequences, which typically contain more than 3,000 frames. For the LeRF experiment, we used the pre-trained 3D-GS models provided by the authors, which were trained using 20–50 frames per scene.
> Regarding computational cost, our full GS-Net training pipeline requires approximately 1 day on 8×A100 GPUs for the ScanNet experiment, and about 12 hours on 8×A100 GPUs for the LeRF experiment. The peak GPU memory usage observed on a single GPU during training was 32 GB for a batch size of one.

---

### Official Review · Reviewer_wYC5 · 2025-10-28

**Soundness:** 2
**Presentation:** 3
**Contribution:** 2
**Rating:** 4
**Confidence:** 4

**Summary:**

This paper presents Quantile Rendering (Q-Render), a novel and efficient rendering algorithm for high-dimensional features in 3D Gaussian Splatting (3D-GS). Traditional 3D-GS-based rendering densely accumulates all Gaussians intersecting each ray, which becomes computationally prohibitive when rendering high-dimensional embeddings such as 512-D CLIP features. Q-Render addresses this by sparsely sampling “quantile Gaussians”, those that contribute most significantly to the transmittance change along the ray.

**Strengths:**

1. The authors evaluate on two large-scale benchmarks with extensive ablations, qualitative visualization, and speed analyses.

2. The approach achieves >40× rendering speedup for 512-D features while improving mIoU—impressive for real-world scalability.

3. Bridges 2D foundation models (CLIP, SAM) with 3D Gaussian representations—a timely and valuable direction for the ICLR community.

**Weaknesses:**

1. The quantile sampling justification is intuitive but lacks a quantitative analysis of approximation error relative to volume rendering.

2. Although ablations are provided, an adaptive or learned K would make the method more robust and generalizable.

3. The paper primarily focuses on indoor datasets (ScanNet, LeRF-OVS). Outdoor or multi-view generalization tests would strengthen the claim of scalability.

4. Only MinkUNet and PTv3 are explored. An analysis of architecture-agnostic behavior would enhance generality.

**Questions:**

1. Could the authors provide a theoretical bound or empirical analysis quantifying the approximation error between Q-Render and full volume rendering?

2. Is it possible to make K adaptive based on the transmittance variance along each ray?

3. How sensitive is the approach to the noise in Gaussian opacity or density estimation?

4. Can Q-Render be applied to RGB image rendering or only to feature maps?

5. What is the memory footprint of Q-Render compared to top-K or compressed-feature methods?

6. Have the authors tested Q-Render for dynamic scenes or time-varying Gaussians?

---

> ### Author Response · Authors · 2025-11-27
> **Theoretical Derivation of Q-Render (1/2)**
>
> **[wYC5, W1] [wYC5, Q1] Theoretical Analysis of Q-Render**
>
>
> We provide a theoretical derivation of Q-Render’s convergence by modeling it as a right Riemann sum. Based on the right endpoint approximation theorem, we prove that the error bound is inversely proportional to the number of samples, ensuring the convergence of Q-Render. Detailed proofs are provided in Appendix C.
>
>
> ## Theoretical Analysis of Quantile Rendering
>
> Based on the definitions provided in Section 4.2 and Algorithm 1 of the manuscript, we provide the theoretical analysis of Quantile Rendering (Q-Render) as an approximation of the volume rendering in 3D-GS using Riemann sums.
>
> ### Preliminaries: Riemann Sums and the Right Rule
>
> Let $f: [a, b] \to \mathbb{R}$ be a bounded function defined on a closed interval $[a, b]$. We define a partition $P$ of the interval into $N$ sub-intervals:
>
> $$
> a = u_0 < u_1 < u_2 < \dots < u_N = b
> $$
>
> The width of the $k$-th sub-interval $[u_{k-1}, u_k]$ is denoted by $\Delta u_k = u_k - u_{k-1}$. A **Riemann Sum** approximates the definite integral of $f$ over $[a, b]$ by summing the areas of rectangles:
>
> $$
> S = \sum_{k=1}^{N} f(u_k^*) \Delta u_k
> $$
>
> where $u_k^*$ is a sample point chosen from the sub-interval $[u_{k-1}, u_k]$.
>
> The **Right Riemann Sum** specifically chooses the right endpoint of each sub-interval as the sample point ($u_k^* = u_k$). Thus, the approximation becomes:
>
> $$
> S_{right} = \sum_{k=1}^{N} f(u_k) \Delta u_k
> $$
>
> When the partition is uniform, $\Delta x = (b-a)/N$, due to the "Right endpoint approximation theorem", the Right Rule converges linearly:
>
> $$
> \left|\int_a^b f(x)\,dx - S_{\mathrm{right}}\right| \le \frac{M(b-a)^2}{2N}
> $$
>
> where $M$ is the maximum value of the absolute value of $\frac{\partial f(x)}{\partial x}$ over the interval.
>
> ### Volume Rendering as an Integral
>
> We observe that the discrete volume rendering process is fundamentally equivalent to a Riemann sum approximation of a continuous path integral. Theoretically, volume rendering assumes that the accumulated transmittance $T$ monotonically decreases from $1$ to $0$ along the ray, creating a bounded integration domain.
>
>
>
> Different neural rendering approaches can be characterized by how they partition this domain:
>
> * **NeRF:** The integral is approximated in the *spatial domain* ($t$). Query points are sampled uniformly or hierarchically (coarse-to-fine) along the ray. The Riemann partition is defined by the distance between adjacent samples ($\delta_i = t_{i+1} - t_i$), where the opacity $\alpha_i$ is derived from the learned volumetric density $\sigma$.
> * **3D-GS:** The integral is evaluated using a set of discrete primitives sorted by depth. Here, the rasterized Gaussians act as the representative samples for intervals along the ray. The rendering equation accumulates the alpha-blended contribution of each intersecting Gaussian, effectively performing a Riemann summation where the step size and weight are determined by the Gaussian's covariance and opacity.
> * **Quantile Rendering (Ours):** Unlike the spatial sampling in NeRF or the dense accumulation in 3D-GS, our method shifts the Riemann partition to the *transmittance domain* ($T$). By sampling intervals of equal probability in transmittance space, we ensure that the summation is performed only over the most significant contributors, offering a more efficient approximation of the underlying integral.

---

> ### Author Response · Authors · 2025-11-27
> **Theoretical Derivation of Q-Render (2/2)**
>
> ### Volume Rendering as a Continuous Integral
>
> Consider a camera ray parameterized by distance $t \in [0,\infty)$. Let $\sigma(t)$ denote the volume density and $c(t)$ the feature (or color) emitted at position $t$.
>
> The transmittance is defined as:
>
> $$
> T(t) = \exp\Big(-\int_0^t \sigma(s)\,ds\Big)
> $$
>
> Its derivative satisfies:
>
> $$
> \frac{dT}{dt} = -\sigma(t) T(t) \quad\Longrightarrow\quad -\frac{dT}{dt}\,dt = \sigma(t)T(t)\,dt
> $$
>
> The continuous volume rendering integral is:
>
> $$
> C_{\mathrm{vol}} = \int_{0}^{\infty} c(t)\,\sigma(t)T(t)\,dt
> $$
>
> **Change of Variables from spatial distance $t$ to Transmittance $T$**
>
> The Quantile render operates by analyzing the change in Transmittance $T$ rather than spatial distance $t$. We can reformulate the volume rendering integral by performing a change of variables from spatial distance $t$ to transmittance $u = T(t)$.
>
>
>
> Following previous works, $u$ and $t$ are noted as:
> * When $t=0$, $u=1$ (Ray starts with full transmittance starting from $1$).
> * When $t \to \infty$, $u \to 0$ (Transmittance converges to $0$).
>
> Since $T$ is strictly decreasing from $1$ to $0$, the substitution $u = T(t)$ and $du = -\sigma(t)T(t)\,dt$ transforms the integral into:
>
> $$
> C_{\mathrm{vol}} = \int_{u=1}^{0} c(u)\,(-du) = \int_{0}^{1} c(u)\,du
> $$
>
> Thus, volume rendering is exactly the integral of a function $c(u)$ over the transmittance interval $[0,1]$:
>
> $$
> C_{\mathrm{vol}} = \int_0^1 c(u)\,du
> $$
>
> ### Quantile Rendering as a Right Riemann Sum
>
> As described in Section 4.2 of the manuscript, the Quantile Rendering algorithm partitions the Transmittance $T \in [0, 1]$ into $K+1$ evenly distributed segments. The algorithm selects a specific "Quantile Gaussian" whenever the accumulated transmittance drops.
>
> In detail, the Quantile Rendering partitions the transmittance domain $[0,1]$ into:
>
> $$
> 0 = u_{K+1} < u_{K} < \cdots < u_1  < u_0 = 1
> $$
>
> For each interval $[u_{k-1}, u_k]$, Q-render selects the Gaussian whose transmittance crossing corresponds to the right endpoint $u_k$. Thus the algorithm evaluates $c(u)$ at $u_k$. Therefore, Q-render implements exactly the Right Riemann Sum for the volume integral:
>
> $$
> C_{Q}^{\mathrm{right}} = \sum_{k=1}^{K+1} c(u_k)\,\Delta u
> $$
>
> This replaces the dense per-Gaussian accumulation with a sparse, quantile-driven sampling strategy.
>
> ### Approximation Error of Quantile Rendering
>
> Following established theory, the integrand $c(u)$ is differentiable and satisfies $|c'(u)| \le M$ for all $u\in[0,1]$. Applying the classical Right Rule error bound with $a=0$, $b=1$, and $N=K+1$ gives:
>
> $$
> \big| C_{\mathrm{vol}} - C_Q^{\mathrm{right}} \big| \le \frac{M}{2(K+1)} \le \frac{M}{2K}
> $$
>
> Thus, the quantile approximation converges at a rate of $O(1/K)$.
>
> ### Convergence of Q-render with Transmittance Normalization
>
> Q-render includes a final normalization step to correct for residual transmittance:
>
> $$
> \widetilde C_Q = \frac{C_Q^{\mathrm{right}}}{1 - T_Q}
> $$
>
> where $T_Q$ is the remaining transmittance after processing all $K$ quantile Gaussians. Since each quantile removes at least $\Delta u$, we have $T_Q \le \Delta u = \frac{1}{K+1}$. Hence the normalization factor satisfies:
>
> $$
> \frac{1}{1 - T_Q} \le \frac{1}{1 - \frac{1}{K+1}} = \frac{K+1}{K}
> $$
>
> Combining this with the previous error bound yields:
>
> $$
> \big|C_{\mathrm{vol}} - \widetilde C_Q \big| \le \frac{K+1}{K} \cdot \frac{M}{2(K+1)} = \frac{M}{2K}
> $$
>
> **Final Theorem**
>
> Under the assumptions that $c$ is differentiable and $|c'(u)|\le M$, Q-render converges to volume rendering with rate:
>
> $$
> \boxed{
> \big|C_{\mathrm{vol}} - \widetilde C_Q \big|
> \le \frac{M}{2K}
> }
> $$
>
> and the approximation error vanishes linearly as $K\to\infty$.

---

> ### Author Response · Authors · 2025-11-27
> **Reviewer wYC5 (1/4)**
>
> **[wYC5, W2] [wYC5, Q2] Adaptive Gaussian sampling**
>
> We further implement two adaptive variants of Q-Render to alleviate its sensitivity to the choice of K. For more model details, please refer to Appendix E.1.
>
> **1)  Learned-K**
>
> In this variant, we train the model with three candidate K values: 10, 20, and 40. Inspired by the mIoU-prediction head in SAM2, we introduce an additional **similarity-prediction head** that estimates the expected similarity between the predicted features and the target features for each candidate K. This head takes the transmittance distribution as input and outputs the predicted similarity scores. During inference, the model evaluates these similarity scores and dynamically selects the K that yields the highest expected similarity. This enables the model to adaptively select the optimal K per input without relying on a manually fixed value.
>
> **2) Stratified-selection**
>
> In the second variant, we compute the mean and standard deviation of the transmittance values along the depth dimension. Instead of uniformly partitioning transmittance for sampling, we sample based on the **z-score under a Gaussian distribution** parameterized by the estimated mean and standard deviation. This allows the model to allocate more samples at depths where objects are more likely to contribute, analogous to the **stratified sampling strategy in NeRF**. The resulting adaptive sampling improves robustness to scene-dependent transmittance distributions.
>
> We compare adaptive variants against Q-Render using a voxel size of 0.5 and K=40. As shown in Table 6, ‘Learned-K’ slightly underperforms due to optimization overhead, while ‘stratified-K’ performs on par with Q-Render. More importantly, both variants exhibit nearly half the FPS of Q-Render due to the necessity of two-pass rendering. Given the lack of performance gain despite the high computational cost, we adopt the original Q-Render for its efficiency and effectiveness.
>
> | Method         | FPS    | 19 mIoU | 19 mAcc | 15 mIoU | 15 mAcc | 10 mIoU | 10 mAcc |
> |----------------|--------|-------------|-------------|-------------|-------------|-------------|-------------|
> | Learned-K      | 14.31  | 40.14  | 49.16  | 43.85  | 55.60 | 55.84 | 68.53 |
> | Stratified-K   | 15.14  | **41.30**   | 49.52       | **44.67**   | **55.78**   | 56.66       | 70.10       |
> | Q-Render (ours)| **32.17** | 41.12    | **49.72**   | 44.40       | 55.32       | **57.05**   | **71.46**   |
>
> **Table 8: Open vocabulary 3D semantic segmentation performances in the ScanNet dataset.**

---

> ### Author Response · Authors · 2025-11-27
> **Reviewer wYC5 (2/4)**
>
> **[wYC5, W3] Outdoor experiments**
>
> To the best of our knowledge, there are currently no available benchmarks that jointly cover multi-view and outdoor scenes for open-vocabulary segmentation (OVS). To address this gap and enable comparison in outdoor settings, we manually annotated 3 scenes from MipNeRF360 outdoor scenes. Specifically, we first use SAM2 to obtain an initial mask corresponding to the queried object and then manually refine the segmentation by providing additional positive and negative point prompts. This process allows for reliable ground-truth masks for evaluating outdoor multi-view OVS performance across all baselines.
> We compare Q-Render with the previous state-of-the-art method, Dr.Splat~\citep{dr_splat}, on the manually annotated MipNeRF360 benchmark. As shown in Table9 and Figure 10, Q-Render consistently outperforms Dr.Splat, in line with our findings in the main paper. This demonstrates that our pipeline generalizes well to outdoor scenarios as well. For more details, please refer to Appendix E.2. We will publicly release the new MipNeRF360-based benchmark for reproducibility.
>
> | Scene    | Dr.Splat mIoU | Dr.Splat mAcc | GS-Mink mIoU | GS-Mink mAcc |
> |----------|----------------|----------------|---------------|---------------|
> | bicycle  | 0.2112         | 0.3012         | **0.2236**    | **0.3165**    |
> | garden   | 0.5543         | 0.6114         | **0.6721**    | **0.7813**    |
> | treehill | **0.2122**     | **0.2713**     | 0.2063        | 0.2585        |
> | average  | 0.3359         | 0.3946         | **0.3673**    | **0.4521**    |
>
> **Table 7: Comparison between Dr.Splat and GS-Mink on MipNeRF360 benchmark.**
>
> **[wYC5, W4] More network architecture**
>
> We have incorporated two additional networks, PointNet++ and PointNext, to further evaluate the architectural behavior of our GS-Net. PointNet++ introduces a hierarchical structure to capture local geometric features at varying metric scales, addressing the lack of local context in the original architecture. PointNext further improves upon PointNet++ through advanced training strategies and architectural refinements.
>
> As shown in Table 10, however, both PointNet++ and PointNext exhibit inferior performance compared to Mink and PTv3. We conjecture that point-based networks struggle to capture local geometries in regions with densely clustered Gaussians, as they rely on k-nearest neighbor (k-NN) search. In such dense regions, k-NN limits the receptive field, hindering the aggregation of broader context. In contrast, voxel-based networks are inherently more robust to varying point densities, demonstrating stronger performance given that 3D Gaussians tend to be densely concentrated around object surfaces.
>
> | 3D Arch. | Rendering | mIoU19 | mAcc19 | mIoU15 | mAcc15 | mIoU10 | mAcc10 |
> | :--- | :---: | :---: | :---: | :---: | :---: | :---: | :---: |
> | **PointNet++** | Q-render | 39.42 | 50.62 | 42.88 | 56.66 | 52.21 | 69.14 |
> | **PointNeXT** | Q-render | 37.89 | 45.80 | 41.15 | 52.09 | 51.57 | 63.83 |
> | **PTv3** | Q-render | 48.99 | 60.36 | 52.39 | 66.05 | 64.95 | 79.34 |
> | **MinkUNet** | Q-render | 50.75 | 62.00 | 53.54 | 66.39 | 64.95 | 79.34 |
>
> **Table 10. Comparison of various 3D network backbones to GS-Net.**

---

> ### Author Response · Authors · 2025-11-27
> **Reviewer wYC5 (3/4)**
>
> **[wYC5, Q3] Sensitivity to noises in Gaussian opacity**
>
> We also investigate the sensitivity of GS-Net to perturbations applied to the input 3D-GS representation. Specifically, on top of the predicted opacity, we inject Gaussian noise with varying scales into the Gaussian opacity values and evaluate the resulting segmentation performance.
>
> As shown in Table 10, GS-Net remains highly robust under mild noise levels: performance degradation at noise scales of 0.25 and 0.5 is negligible, demonstrating that Q-Render retains sufficient stability even when the opacity field is moderately corrupted. However, as the noise scale increases beyond 1.0, we observe a marginal drop in both mIoU and mAcc, and the performance eventually collapses at extreme noise levels such as 4.0. These results indicate that while Q-Render is resilient to realistic perturbations in the opacity field, strong distortions that heavily corrupt the underlying geometry inevitably degrade performance. Overall, this experiment suggests that our pipeline maintains robust performance under practical levels of noise in 3D-GS inputs. For more experimental details, please refer to Appendix E.5.
> | Noise Scale | mIoU19 | mAcc19 |
> |-------------|--------|--------|
> | 0           | 50.75  | 62.00  |
> | 0.25        | 50.13  | 60.94  |
> | 0.5         | 50.18  | 60.88  |
> | 1           | 47.13  | 57.10  |
> | 2           | 38.13  | 44.11  |
> | 4           | 16.12  | 32.19  |
> **Table 10: Robustness of Q-Render to Gaussian noise added to opacity.**
>
> **[wYC5, Q5] Memory footprint of LangSplat, OpenGaussian, Dr.Splat (top-K), and Quantile Render.**
>
> We also compare the VRAM requirements of each baseline on `scene0000_00` from ScanNet (100 frames). As summarized in Table 11, LangSplat and OpenGaussian require less memory due to their low-dimensional feature representations (3D and 6D). However, this compression inevitably introduces information loss, which correlates with their reduced segmentation performance. Dr.Splat relies on 512D features and stores per-view Gaussian visibility masks, which causes memory usage to grow proportionally with the number of frames and leads to more than 61 GB of peak consumption. Q-Render also uses 512D features, and its cache-free per-ray accumulation avoids this overhead, reducing the peak memory to 27.18 GB at \(K = 40\) while preserving high-dimensional semantic capacity. For more details, please refer to Table 11.
>
> | Method             | Feature Dim. | Peak Memory (GB) |
> |--------------------|--------------|------------------|
> | LangSplat          | 3            | 7.18             |
> | OpenGaussian       | 6            | 16.13            |
> | Dr.Splat           | 512          | 61.13            |
> | Q-Render (K=40)    | 512          | 27.18            |
>
> **Table 11. Peak inference memory on `scene0000_00` (ScanNet, 100 frames)**

---

> ### Author Response · Authors · 2025-11-27
> **Reviewer wYC5 (4/4)**
>
> **[wYC5, Q4] [wYC5, Q6] Applying Q-render for RGB rendering.**
>
> Since our Q-Render is not restricted to CLIP features, it could also be applied to RGB image rendering. Using this setup, we quantify how much information is lost when replacing V-Render with Q-Render. To isolate the behavior of the renderer itself, we directly apply Q-Render to pre-trained 3D-GS models without any fine-tuning. As shown in Fig.10, Q-Render exhibits only a very slight drop in PSNR compared to V-Render, indicating that it provides an effective approximation of the original renderer even in the RGB domain.
> This suggests that Q-Render is broadly applicable beyond our semantic feature rendering setup. In particular, we believe Q-Render can be seamlessly integrated into various 3D-GS variants—including dynamic or time-varying Gaussians—since these models also assume fixed Gaussian positions at each rendering step. For more details, please refer to Appendix E.6.

---

### Official Review · Reviewer_J4U2 · 2025-11-01

**Soundness:** 3
**Presentation:** 1
**Contribution:** 3
**Rating:** 4
**Confidence:** 4

**Summary:**

This paper proposed quantile rendering to accelerate the rasterization of 3D Gaussians. Specifically, the proposed method selects a subset of critical 3D Gaussians that have a significant influence on the final rendering results and skips the rest. The experiments on 3D open vocabulary segmentation show that the proposed method can speed up the rendering while achieving state-of-the-art performance.

**Strengths:**

The idea is interesting and well-motivated.
The performances shown in the experiments are good.

**Weaknesses:**

1. Paper presentation: In the abstract, the GS-Net and open-vocabulary 3D semantic segmentation are mentioned, without introducing their relationship with the Q-Render, making it hard to follow. The topic of the paper is unclear. If the proposed method is designed for general high-dimensional feature rendering, why is it only evaluated on the 3D open-vocabulary semantic segmentation task? If the method is specifically designed for 3D open-vocabulary semantic segmentation, then there is a lack of introduction to the specific problem.
2. Unreliable experiments: The final method used for 3D open-vocabulary semantic segmentation consists of two parts: the per-Gaussian feature extracted by GS-Net and the Q-Rendering procedure. It is hard to justify the separate contribution of the two parts to the performance gain in Tab. 3. To my understanding, as Q-Render only requires Gaussians with high-dimensional features as input, it is not hard to disentangle the two parts: (1) Replace the rendering algorithm of the baselines using Q-Render and compare their performance. (2) Replace the feature extraction model of the baselines using the GS-Net and compare their performance.
3. Minor typos: in L.199, the reference line 6 of Algorithm 1 should be enclosed in parentheses.

**Questions:**

Please see weaknesses.

---

> ### Author Response · Authors · 2025-11-27
> **Reviewer J4U2**
>
> **[J4U2, W1] Reorganizing Abstract and Introduction to reveal the OVS task**
>
> We appreciate your valuable suggestions regarding the paper's presentation. Agreeing with your points, we have updated the Title, Abstract, and Introduction to clearly emphasize that our pipeline is designed to address the OVS task and its challenges. We hope this revised version adequately resolves your concerns, as we believe clear presentation is essential for communicating our contributions.
>
> **[J4U2, W2] Supplementing ablation studies**
>
> To better understand which components of our method contribute most to the performance improvement, we conduct an ablation by modifying each part of the pipeline individually and comparing the results against OpenGaussian. Specifically, we introduce two hybrid models: one that uses the GS-Net feature extractor while retaining the original rendering module of OpenGaussian, and another that keeps the OpenGaussian feature extractor but replaces the renderer with Q-Render.
>
> As shown in Table 9, incorporating Q-Render yields a consistent performance gain for both OpenGaussian and GS-Net. Moreover, replacing the OpenGaussian codebooks with GS-Net features leads to a substantial improvement. We conjecture that preserving high-dimensional features is a key factor, as OpenGaussian encodes only 6-dimensional attributes, whereas GS-Net leverages 512-dimensional CLIP features that retain richer semantic information.
>
> | Feature Extractor | Rendering | mIoU19 | mAcc19 | mIoU15 | mAcc15 | mIoU10 | mAcc10 |
> |-------------------|-----------|--------|--------|--------|--------|--------|--------|
> | OpenGaussian      | V-Render  | 22.60  | 34.41  | 24.21  | 37.58  | 34.74  | 51.58  |
> |                   | Q-Render  | 23.10  | 35.55  | 26.18  | 38.12  | 36.13  | 56.12  |
> | GS-Net (Mink)     | V-Render  | 49.02  | 58.75    | 50.41     | 63.65     |  61.04     | 76.21    |
> |                   | Q-Render  | 50.75  | 62.00  | 53.54  | 66.39  | 64.94  | 79.34  |
>
> **Table 9: Comparison of V-Render and Q-Render under different feature extractors**
>
> Along with this ablation study, we have insightful background to introduce the 3D neural networks, such as MinkUNet and PTv3, on top of Q-Render in the introduction section:
>
> > _Moreover, with this integration, Q-Render’s sparse sampling becomes even more advantageous: the inductive bias of the 3D neural network tends to predict spatially smooth Gaussian features, meaning that densely sampling all Gaussians along each ray is unnecessary. Instead, the sparsely selected quantile Gaussians are sufficient to faithfully render high-dimensional feature maps while significantly reducing computational overhead during the rendering process and its backward computation._
>
> With such a reason, we leverage the existing 3D neural networks to our Q-Render. Additional ablation studies regarding the 3D neural networks are provided in Table 10 of the Appendix.
>
> **[J4U2, W3] Typo fix**
>
> Thanks for finding the typo. We have updated in the revised version.

---

> ### Comment · Reviewer_J4U2 · 2025-11-28
> **Thank You for the Rebuttal**
>
> Thank you for your response and additional results, which addresses my main concern. I therefore upgrade my rating to 6.
>
> update: I didn't find a place to update the rating, will update when I can do so.

---

### Official Review · Reviewer_Fhos · 2025-11-01

**Soundness:** 3
**Presentation:** 3
**Contribution:** 3
**Rating:** 6
**Confidence:** 3

**Summary:**

This paper presents Q-Render, an efficient rendering algorithm for high-dimensional feature rendering in 3D Gaussian Splatting. Q-Render sparsely samples dominant Gaussians through transmittance change analysis. The authors integrate Q-Render with a 3D neural network (GS-Net) and evaluate it on two open-vocabulary 3D semantic segmentation benchmarks: ScanNet and LeRF-OVS, demonstrating superior performance.

**Strengths:**

1. The paper proposes a quantile sampling strategy that identifies critical Gaussians through transmittance analysis, which is theoretically motivated and practically effective.

2.  The method demonstrates superior performance and efficiency on both ScanNet and LeRF-OVS open-vocabulary 3D semantic segmentation benchmarks, achieving ~43.7× speed gains when rendering 512-D feature maps.

**Weaknesses:**

1. The paper lacks theoretical analysis of Q-Render's approximation error to volume rendering or convergence guarantees. The mathematical justification for the normalization operation (line 20 in Algorithm 1) is insufficient.

2. While K significantly impacts performance, the paper lacks an adaptive strategy for selecting K. Why is uniform partitioning (k+1)/(K+1) chosen? Have adaptive thresholds been considered?

3. The paper does not sufficiently analyze when Q-Render might fail or how it performs on scenes with non-uniform transmittance distributions.

4. In Table 2, the authors "reproduce" baseline results but use different training and evaluation setups, which may not provide a fair comparison.

5.  The necessity of de-voxelization is not independently validated. The paper lacks failure case analysis showing scenarios where Q-Render underperforms.

**Questions:**

1. Can you provide an error bound for Q-Render's approximation to volume rendering? What are the theoretical guarantees for the normalization step?

2.  Is the optimal value of K related to scene complexity and feature dimensionality? Can you design an adaptive strategy for K selection?

3. Performance anomaly: Why does Q-Render with K=40 achieve higher mIoU (50.85) than volume rendering (49.02) in Table 6? Does this suggest that sparse sampling has a regularization effect?

4.  How can the information loss from voxelization (Figure 7) be quantified? Are there better voxelization strategies to mitigate this loss?

5. Threshold design: Is uniform partitioning of transmittance thresholds optimal? Have you considered scene-adaptive thresholds based on local transmittance distributions?

6.  How is de-voxelization specifically implemented? Is K fixed during both training and inference? How do you handle regions with very small transmittance changes?

---

> ### Author Response · Authors · 2025-11-27
> **Theoretical Derivation of Q-Render (1/2)**
>
> **[Fhos, W1] [Fhos, Q1] Theoretical Analysis of Q-Render**
>
>
> We provide a theoretical derivation of Q-Render’s convergence by modeling it as a right Riemann sum. Based on the right endpoint approximation theorem, we prove that the error bound is inversely proportional to the number of samples, ensuring the convergence of Q-Render. Detailed proofs are provided in Appendix C.
>
>
> ## Theoretical Analysis of Quantile Rendering
>
> Based on the definitions provided in Section 4.2 and Algorithm 1 of the manuscript, we provide the theoretical analysis of Quantile Rendering (Q-Render) as an approximation of the volume rendering in 3D-GS using Riemann sums.
>
> ### Preliminaries: Riemann Sums and the Right Rule
>
> Let $f: [a, b] \to \mathbb{R}$ be a bounded function defined on a closed interval $[a, b]$. We define a partition $P$ of the interval into $N$ sub-intervals:
>
> $$
> a = u_0 < u_1 < u_2 < \dots < u_N = b
> $$
>
> The width of the $k$-th sub-interval $[u_{k-1}, u_k]$ is denoted by $\Delta u_k = u_k - u_{k-1}$. A **Riemann Sum** approximates the definite integral of $f$ over $[a, b]$ by summing the areas of rectangles:
>
> $$
> S = \sum_{k=1}^{N} f(u_k^*) \Delta u_k
> $$
>
> where $u_k^*$ is a sample point chosen from the sub-interval $[u_{k-1}, u_k]$.
>
> The **Right Riemann Sum** specifically chooses the right endpoint of each sub-interval as the sample point ($u_k^* = u_k$). Thus, the approximation becomes:
>
> $$
> S_{right} = \sum_{k=1}^{N} f(u_k) \Delta u_k
> $$
>
> When the partition is uniform, $\Delta x = (b-a)/N$, due to the "Right endpoint approximation theorem", the Right Rule converges linearly:
>
> $$
> \left|\int_a^b f(x)\,dx - S_{\mathrm{right}}\right| \le \frac{M(b-a)^2}{2N}
> $$
>
> where $M$ is the maximum value of the absolute value of $\frac{\partial f(x)}{\partial x}$ over the interval.
>
> ### Volume Rendering as an Integral
>
> We observe that the discrete volume rendering process is fundamentally equivalent to a Riemann sum approximation of a continuous path integral. Theoretically, volume rendering assumes that the accumulated transmittance $T$ monotonically decreases from $1$ to $0$ along the ray, creating a bounded integration domain.
>
>
>
> Different neural rendering approaches can be characterized by how they partition this domain:
>
> * **NeRF:** The integral is approximated in the *spatial domain* ($t$). Query points are sampled uniformly or hierarchically (coarse-to-fine) along the ray. The Riemann partition is defined by the distance between adjacent samples ($\delta_i = t_{i+1} - t_i$), where the opacity $\alpha_i$ is derived from the learned volumetric density $\sigma$.
> * **3D-GS:** The integral is evaluated using a set of discrete primitives sorted by depth. Here, the rasterized Gaussians act as the representative samples for intervals along the ray. The rendering equation accumulates the alpha-blended contribution of each intersecting Gaussian, effectively performing a Riemann summation where the step size and weight are determined by the Gaussian's covariance and opacity.
> * **Quantile Rendering (Ours):** Unlike the spatial sampling in NeRF or the dense accumulation in 3D-GS, our method shifts the Riemann partition to the *transmittance domain* ($T$). By sampling intervals of equal probability in transmittance space, we ensure that the summation is performed only over the most significant contributors, offering a more efficient approximation of the underlying integral.

---

> ### Author Response · Authors · 2025-11-27
> **Theoretical Derivation of Q-Render (2/2)**
>
> ### Volume Rendering as a Continuous Integral
>
> Consider a camera ray parameterized by distance $t \in [0,\infty)$. Let $\sigma(t)$ denote the volume density and $c(t)$ the feature (or color) emitted at position $t$.
>
> The transmittance is defined as:
>
> $$
> T(t) = \exp\Big(-\int_0^t \sigma(s)\,ds\Big)
> $$
>
> Its derivative satisfies:
>
> $$
> \frac{dT}{dt} = -\sigma(t) T(t) \quad\Longrightarrow\quad -\frac{dT}{dt}\,dt = \sigma(t)T(t)\,dt
> $$
>
> The continuous volume rendering integral is:
>
> $$
> C_{\mathrm{vol}} = \int_{0}^{\infty} c(t)\,\sigma(t)T(t)\,dt
> $$
>
> **Change of Variables from spatial distance $t$ to Transmittance $T$**
>
> The Quantile render operates by analyzing the change in Transmittance $T$ rather than spatial distance $t$. We can reformulate the volume rendering integral by performing a change of variables from spatial distance $t$ to transmittance $u = T(t)$.
>
>
>
> Following previous works, $u$ and $t$ are noted as:
> * When $t=0$, $u=1$ (Ray starts with full transmittance starting from $1$).
> * When $t \to \infty$, $u \to 0$ (Transmittance converges to $0$).
>
> Since $T$ is strictly decreasing from $1$ to $0$, the substitution $u = T(t)$ and $du = -\sigma(t)T(t)\,dt$ transforms the integral into:
>
> $$
> C_{\mathrm{vol}} = \int_{u=1}^{0} c(u)\,(-du) = \int_{0}^{1} c(u)\,du
> $$
>
> Thus, volume rendering is exactly the integral of a function $c(u)$ over the transmittance interval $[0,1]$:
>
> $$
> C_{\mathrm{vol}} = \int_0^1 c(u)\,du
> $$
>
> ### Quantile Rendering as a Right Riemann Sum
>
> As described in Section 4.2 of the manuscript, the Quantile Rendering algorithm partitions the Transmittance $T \in [0, 1]$ into $K+1$ evenly distributed segments. The algorithm selects a specific "Quantile Gaussian" whenever the accumulated transmittance drops.
>
> In detail, the Quantile Rendering partitions the transmittance domain $[0,1]$ into:
>
> $$
> 0 = u_{K+1} < u_{K} < \cdots < u_1  < u_0 = 1
> $$
>
> For each interval $[u_{k-1}, u_k]$, Q-render selects the Gaussian whose transmittance crossing corresponds to the right endpoint $u_k$. Thus the algorithm evaluates $c(u)$ at $u_k$. Therefore, Q-render implements exactly the Right Riemann Sum for the volume integral:
>
> $$
> C_{Q}^{\mathrm{right}} = \sum_{k=1}^{K+1} c(u_k)\,\Delta u
> $$
>
> This replaces the dense per-Gaussian accumulation with a sparse, quantile-driven sampling strategy.
>
> ### Approximation Error of Quantile Rendering
>
> Following established theory, the integrand $c(u)$ is differentiable and satisfies $|c'(u)| \le M$ for all $u\in[0,1]$. Applying the classical Right Rule error bound with $a=0$, $b=1$, and $N=K+1$ gives:
>
> $$
> \big| C_{\mathrm{vol}} - C_Q^{\mathrm{right}} \big| \le \frac{M}{2(K+1)} \le \frac{M}{2K}
> $$
>
> Thus, the quantile approximation converges at a rate of $O(1/K)$.
>
> ### Convergence of Q-render with Transmittance Normalization
>
> Q-render includes a final normalization step to correct for residual transmittance:
>
> $$
> \widetilde C_Q = \frac{C_Q^{\mathrm{right}}}{1 - T_Q}
> $$
>
> where $T_Q$ is the remaining transmittance after processing all $K$ quantile Gaussians. Since each quantile removes at least $\Delta u$, we have $T_Q \le \Delta u = \frac{1}{K+1}$. Hence the normalization factor satisfies:
>
> $$
> \frac{1}{1 - T_Q} \le \frac{1}{1 - \frac{1}{K+1}} = \frac{K+1}{K}
> $$
>
> Combining this with the previous error bound yields:
>
> $$
> \big|C_{\mathrm{vol}} - \widetilde C_Q \big| \le \frac{K+1}{K} \cdot \frac{M}{2(K+1)} = \frac{M}{2K}
> $$
>
> **Final Theorem**
>
> Under the assumptions that $c$ is differentiable and $|c'(u)|\le M$, Q-render converges to volume rendering with rate:
>
> $$
> \boxed{
> \big|C_{\mathrm{vol}} - \widetilde C_Q \big|
> \le \frac{M}{2K}
> }
> $$
>
> and the approximation error vanishes linearly as $K\to\infty$.

---

> ### Author Response · Authors · 2025-11-27
> **Reviewer Fhos (1/3)**
>
> **[Fhos, W1] [Fhos, Q1] Theoretical Analysis of Q-Render**
>
> **Transmittance-based normalization**
>
> For transmittance-based normalization, we first note that our task uses OpenCLIP, which outputs unit-norm (L2-normalized) features. Thus, regardless of the normalization applied within Q-Render, the final predicted features are always normalized to have unit norm. In other words, even if we apply transmittance-based normalization during rendering, the final rendered feature is subsequently normalized to a unit vector when used for CLIP similarity computation, resulting in the same output.
>
> Nevertheless, applying our normalization is still beneficial for visualizing the rendered feature maps, as it ensures that the feature magnitudes remain consistent across pixels. Without this normalization, the rendered feature map often exhibits unstable magnitude patterns—even though the normalized features may remain similar to the target text embedding. Therefore, we include normalization primarily to improve model interpretability.
>
>
> **[Fhos, W2] [Fhos, Q2] [Fhos, Q5] Adaptive Gaussian sampling strategy**
>
> We further implement two adaptive variants of Q-Render to alleviate its sensitivity to the choice of K. For more model details, please refer to Appendix E.1.
>
> **1)  Learned-K**
>
> In this variant, we train the model with three candidate K values: 10, 20, and 40. Inspired by the mIoU-prediction head in SAM2, we introduce an additional **similarity-prediction head** that estimates the expected similarity between the predicted features and the target features for each candidate K. This head takes the transmittance distribution as input and outputs the predicted similarity scores. During inference, the model evaluates these similarity scores and dynamically selects the K that yields the highest expected similarity. This enables the model to adaptively select the optimal K per input without relying on a manually fixed value.
>
> **2) Stratified-selection**
>
> In the second variant, we compute the mean and standard deviation of the transmittance values along the depth dimension. Instead of uniformly partitioning transmittance for sampling, we sample based on the **z-score under a Gaussian distribution** parameterized by the estimated mean and standard deviation. This allows the model to allocate more samples at depths where objects are more likely to contribute, analogous to the **stratified sampling strategy in NeRF**. The resulting adaptive sampling improves robustness to scene-dependent transmittance distributions.
>
> We compare adaptive variants against Q-Render using a voxel size of 0.5 and K=40. As shown in Table 6, ‘Learned-K’ slightly underperforms due to optimization overhead, while ‘stratified-K’ performs on par with Q-Render. More importantly, both variants exhibit nearly half the FPS of Q-Render due to the necessity of two-pass rendering. Given the lack of performance gain despite the high computational cost, we adopt the original Q-Render for its efficiency and effectiveness.
>
> | Method         | FPS    | 19 mIoU | 19 mAcc | 15 mIoU | 15 mAcc | 10 mIoU | 10 mAcc |
> |----------------|--------|-------------|-------------|-------------|-------------|-------------|-------------|
> | Learned-K      | 14.31  | 40.14  | 49.16  | 43.85  | 55.60 | 55.84 | 68.53 |
> | Stratified-K   | 15.14  | **41.30**   | 49.52       | **44.67**   | **55.78**   | 56.66       | 70.10       |
> | Q-Render (ours)| **32.17** | 41.12    | **49.72**   | 44.40       | 55.32       | **57.05**   | **71.46**   |
> **Table 8: Open vocabulary 3D semantic segmentation performances in the ScanNet dataset.**

---

> ### Author Response · Authors · 2025-11-27
> **Reviewer Fhos (2/3)**
>
> **[Fhos, W3] [Fhos, W5] Failure cases of Q-Render.**
>
> We have revised the paper to include a detailed discussion on limitations and future work, focusing on three key aspects: the sensitivity of the hyperparameter $K$, the dependence on the quality of input 3D Gaussians, and the dependence on 3D architecture. For more information, please refer to Section 6.
>
> **1. Sensitivity to Dynamic $K$ Selection**
>
> We analyzed the impact of varying the number of samples $K$ during inference using a model trained with $K=40$. As shown in the table below, performance drops when the inference $K$ deviates from the training configuration, indicating the need for consistent sampling. While we explored adaptive strategies (detailed in Appendix), they currently incur high computational costs. Therefore, we leave the development of efficient adaptive sampling for future work.
>
> | $K$ (Inference) | 5 | 10 | 20 | 40 (Train) | 50 |
> | :--- | :---: | :---: | :---: | :---: | :---: |
> | **mIoU** | 39.16 | 42.18 | 44.94 | **45.81** | 45.71 |
> | **mAcc** | 48.43 | 53.94 | 56.14 | **56.87** | 56.94 |
>
> **Table 1: Performance change on the number of $K$. The model is trained with a fixed $K=40$.**
>
> **2. Dependence on 3D-GS Quality**
>
> We acknowledge that our framework depends on the quality of the input 3D Gaussians. To validate this, we compare GS-Net v1 (standard 3D-GS) with GS-Net v2 (depth-supervised 3D-GS); note that the official 3D-GS implementation did not support depth supervision during the GS-Net v1 project and the current official 3D-GS implementation uses depth supervision as default. The table below demonstrates that geometric accuracy significantly impacts segmentation performance. We anticipate that integrating future advancements in generalized 3D-GS, such as DepthSplat [1], WorldMirror [2], and DepthAnything3 [3] will further enhance the scalability and performance of our method.
>
> | Method | Training 3D-GS | mIoU19 / mAcc19 | mIoU15 / mAcc15 | mIoU10 / mAcc10 |
> | :--- | :--- | :---: | :---: | :---: |
> | GS-Net v1 | Rendering Only | 28.42 / 38.85 | 31.02 / 44.02 | 42.58 / 57.92 |
> | **GS-Net v2 (Ours)** | Rendering + Depth | **50.75 / 62.00** | **53.54 / 66.39** | **64.94 / 79.34** |
> **Table 2: Comparison of GS-Net performance across different input 3D-GS models.**
>
> **3. Dependence on 3D Network Architecture.**
> As reported in Appendix E.3, we observe that performance highly depends on the choice of backbone network. In particular, MinkowskiNet  and PTv3 exhibit strong sensitivity to the voxel grid resolution, as shown in Table 4. This suggests that developing a more efficient and Gaussian-aware architecture could further improve the performance of our GS-Net framework. Potential directions include introducing Gaussian-friendly operators, reducing or eliminating voxelization, and designing modules that are robust to noise in Gaussian parameters.
>
>
> **[Fhos, W4] Unfair comparison with previous baselines**
>
>
> One of the previous works, OpenGaussian, disables the densification stage and sets Gaussian positions (\mu) to be the same as the locations of the annotated ground truth points during training for evaluation convenience. However, as emphasized in the original 3D-GS paper, updating Gaussian locations and applying densification / pruning steps are both crucial for achieving reliable rendering quality. This problem has been already addressed in Dr.Splat, and we follow their evaluation protocol. As described in Appendix B.1, we derive Gaussian-level semantic labels by post-processing the trained 3D-GS model, mapping each Gaussian to the point-level ScanNet labels—following a procedure similar to Dr.Splat. However, the official implementation by Dr.Splat does not provide the pre-trained 3D Gaussians or the training / evaluation pipeline in the ScanNet dataset. So, we re-evaluate the previous methods including LangSplat, OpenGaussian, Dr.Splat on top of the same 3D Gaussians and provide the results in the manuscript and Appendix.

---

> ### Author Response · Authors · 2025-11-27
> **Reviewer Fhos (3/3)**
>
> **[Fhos, W5] [Fhos, Q4] Quantifying the information loss during voxelization and de-voxelization**
>
>
> In Section 5.3 and Figure 7, we assess how different voxelization and de-voxelization strategies affect rendering quality. To more directly quantify information loss, we design the experiment such that the ability to reconstruct the original features from voxelized representations serves as an indicator of how much information is discarded during voxelization. A lower reconstruction quality therefore indicates more information loss.
> The column labeled “Voxel Raster” in Figure 7 corresponds to the setting where de-voxelization is completely omitted. To assess the necessity of the de-voxelization step, we preserved the original volumetric rendering pipeline during evaluation. For this configuration, we adopted the sparse voxel rasterizer introduced in SVR, which has been shown to achieve rendering quality comparable to 3D-GS. As reported, removing the de-voxelization step results in a significant PSNR drop (17.42 → 10.21), clearly demonstrating that rendering directly from voxelized features causes severe information loss. This substantial degradation highlights the critical role of de-voxelization in maintaining high-fidelity rendering.
> We also acknowledge that GS-Net could further benefit from a more advanced voxelization strategy. However, in this work, we deliberately focus on developing a generalizable OVS framework with real-time rendering, independent of architectural enhancements to the underlying model. While methods such as FPT[4] provide stronger voxelization designs, we leave the integration and exploration of such strategies for future work.
>
> **[Fhos, Q3] Regularization effect of selecting dominant Gaussians**
>
> When sampling more than 40 Gaussians for each ray, we observed a slight performance drop as reported in Figure 6. This phenomenon is also reported in Dr.Splat, where gradients often flow into many non-dominant Gaussians when too many Gaussians are sampled along the ray. These less-informative Gaussians tend to carry irrelevant or noisy features, which can hinder optimization. According to Dr.Splat (Section 4.4), **selecting only the dominant Gaussians based on ray transmittance provides a strong regularization effect**, leading to notable performance improvements over prior baselines. Since Q-Render similarly focuses on dominant Gaussians along the ray, it naturally exhibits similar behavior.
>
> **[Fhos, Q6] Questions regarding the model designs**
>
>
> **De-voxelization implementation.**
> We used the de-voxelization module from MinkowskiNet, which is widely adopted and validated in prior work. For better reproducibility, we have updated three pseudo codes describing details of voxelization, voxel feature composition, and de-voxelization in Appendix F(Listing 1, 2, and 3).
>
>  **Consistency of K.**
>
> We fix the value of K to be the same during both training and inference.
>
> **Behavior under small transmittance variations.**
>
> When transmittance changes are small, the corresponding regions typically represent less informative structures such as floaters. In these cases, Q-Render naturally samples fewer Gaussians, which effectively reduces attention to such low-impact regions during rendering. This behavior contributes to more robust and focused feature aggregation.
>
>
> [1] DepthSplat: Connecting Gaussian Splatting and Depth, Xu et. al., CVPR2025.
>
> [2] WorldMirror: Universal 3D World Reconstruction with Any-Prior Prompting, Lit et. al., arXiv25.
>
> [3] Depth Anything 3: Recovering the Visual Space from Any Views, L:in et. al., arXiv25.
>
> [4] Fast Point Transformer, Park et. al., CVPR2022.

---

### Author Response · Authors · 2025-11-27
**Common Reviewer Concerns**

We thank all reviewers for their constructive and insightful feedback. We have incorporated the suggested improvements through additional experiments, clarification, and revisions to the manuscript. Below, we summarize the common concerns raised across reviewers and address reviewer-specific comments in separate threads. For brevity, we refer to reviewers using their IDs **[Fhos, J4U2, wYC5, hrzW]**, and denote [W] for weaknesses and [Q] for questions.

As noted by the reviewers, Q-Render is considered well-motivated **[Fhos, J4U2, hrzW]**, achieves strong performance **[Fhos, J4U2, wYC5, hrzW]**, and includes informative control experiments **[Ff9Y, nX6A, 5RcK]**. We appreciate these positive assessments. Reviewers also offered several valuable suggestions for further strengthening the work, and we provide our responses to those points below.


We also note that after the submission, we observed performance gains by simply training for more epochs without modifying any hyperparameters. Accordingly, we have updated the corresponding results in the revised manuscript. In particular, we found that the performance on the LeRF dataset improved substantially, as reported in Table 3.

---

### Author Response · Authors · 2025-12-04
**Final Discussion Summary**

Dear Area Chairs,

Our paper introduces Q-Render, a novel sampling strategy that enables efficient rendering of high-dimensional features (e.g., 512-D CLIP) with a **43.7x** speedup while maintaining high segmentation fidelity. The reviewers have highlighted the following strengths:
- **Motivation and Timeliness**: The idea is described as well-motivated (Fhos, J4U2, hrzW) and a "timely and valuable direction" for bridging 2D foundation models with 3D representations (wYC5).
- **Performance and Efficiency**: The method demonstrates "superior performance and efficiency" (Fhos) and is "impressive for real-world scalability" (wYC5), effectively reducing computational complexity.
- **Experimental Rigor**: The submission includes "extensive ablations, qualitative visualization, and speed analyses" (wYC5) and demonstrates "solid integration & generality" (hrzW).

**Reviewer feedback overview**:

- **Reviewer J4U2** (Rating updated to 6): Explicitly stated "Thank you for your response and additional results, which addresses my main concern. I therefore upgrade my rating to 6."
- **Reviewer Fhos** (Rating 6): Positive assessment maintained.
- **Reviewer wYC5** (Rating 4): Acknowledged strengths; we believe the rebuttal addressed the theoretical and generalization concerns.
- **Reviewer hrzW** (Rating 4): Acknowledged efficiency; we addressed concerns regarding baselines and motivation.

We provided comprehensive responses to all reviewers’ comments, and we believe the main concerns have been successfully addressed:

**For Reviewers Fhos and wYC5**:

- Regarding **"Theoretical Analysis"**: We provided a formal theoretical derivation (Appendix C) modeling Q-Render as a Right Riemann Sum approximation of the continuous volume rendering integral. We proved that the approximation error is bounded by $O(1/K)$, providing the theoretical guarantee requested by the reviewers.
- Regarding **"Adaptive K Strategy"**: We implemented and evaluated two adaptive variants ("Learned-K" and "Stratified-K"). Our results demonstrated that these methods double the computational cost (halving FPS) without significant performance gains, empirically justifying our design choice of a fixed $K$ for optimal efficiency-accuracy trade-off.
- Regarding **"Robustness and Failure Cases"**: We included sensitivity analyses on Gaussian opacity noise and dynamic $K$ selection (Appendix E.5, Section 6), showing the method remains robust under practical noise levels.

**For Reviewer J4U2**:

- Regarding **"Presentation and Scope"**: We revised the Abstract and Introduction to clearly frame the work within the Open-Vocabulary Segmentation (OVS) task context.
- Regarding **"Disentangling Contributions"**: We conducted a thorough ablation study (Table 9) isolating the renderer (Q-Render vs. V-Render) from the feature extractor (GS-Net vs. OpenGaussian). This validated that both Q-Render and GS-Net independently contribute to significant performance gains. **This addressed the reviewer's main concern, leading to a raised score.**

**For Reviewer wYC5**:

- Regarding **"Outdoor Generalization"**: We extended our evaluation to outdoor scenes by manually annotating the MipNeRF360 dataset. We demonstrated that Q-Render consistently outperforms the state-of-the-art (Dr.Splat) in outdoor settings (Appendix E.2).
- Regarding **"Network Architectures"**: We evaluated additional backbones (PointNet++, PointNeXT) alongside MinkUNet and PTv3. Results confirmed that voxel-based architectures are more robust for this task, but our method generalizes across them (Table 10). Additionally, we validated our contribution by replacing the rendering mechanism in OpenGaussian with our Quantile rendering, which yielded improved performance over volume rendering (Table 11).
- Regarding **"RGB Rendering"**: We demonstrated that Q-Render can be applied to RGB rendering with negligible PSNR drop, indicating broad applicability beyond feature rendering. Note that we provide a video demo for the RGB rendering.

**For Reviewer hrzW**:
- Regarding **"Performance Improvement vs. Motivation"**: We clarified that the primary motivation is the substantial rendering speedup (1.4x overall, ~43x on features) which is critical for interactive applications, rather than just mIoU gains.
- Regarding **"Concurrent Work"**: We clarified that the suggested baselines (SIU3R, SegMASt3R) were released within days of or after the ICLR deadline, making comparison infeasible under standard policy.
- Regarding **"Implementation Details"**: We provided the requested details on input views, training time (~1 day on 8x A100s for ScanNet), and peak memory usage.

We thank all reviewers and the Area Chairs for their time, constructive feedback, and thoughtful consideration of our work.

Best regards,
Authors

---

### Meta-Review · Area_Chair_JV2C · 2026-01-04

**Summary:**

This paper receives 1 positive ratings and 3 negative ratings. One reviewer upgraded the rating to positive (4->6) after rebuttal and discussion.

The main concern lies in 1) the lack of theoretical analysis; 2) sensitivity of hyperparameter K; 3) ablation studies on GS-Net and Q-Render; 4) ablation study on the architectural choices.

AC checked all reviews and authors' responses. Concerns on 1), 2) are addressed properly.  However, 3) and 4) further leads AC to a concern on the mismatch between the proposed rendering technique (Q-Render) and the target task (Open-vocabulary segmentation (OVS)). Q-Render is an effective speed-up rendering technique for 3D-GS, especially for 3D-GS with high-dimensional features, yet it is not specific to OVS. It looks like OVS is one task that requires high-dimensional features to work as shown in experiment results. This raises inconsistency between the contribution and the target. If the target is Q-Render, validating its value in high-dimensional 3D-GS across several tasks is necessary and impactful. If the target is OVS, GS-Net is the main focus but is not the main contribution.

Given this mismatch, AC suggests to clarify the main research problem and reshape the manuscript accordingly. The decision for this round is reject.

**Reviewer Concerns:**

Concerns on theoretical analysis, sensitivity of hyperparameter K are addressed. Yet the concern on the focus of this paper still remains.

**Reviewer Scores:**

Reviewer wYC5 may change his/her score because part of his/her concerns are addressed by authors' rebuttal.

---

### Decision · Program_Chairs · 2026-01-26

Reject